# Sublinear Time Low-Rank Approximation of Distance Matrices

**Ainesh Bakshi**
Department of Computer Science
Carnegie Mellon University
Pittsburgh, PA 15213
abakshi@cs.cmu.edu

**David P. Woodruff**
Department of Computer Science
Carnegie Mellon University
Pittsburgh, PA 15213
dwoodruf@cs.cmu.edu

## Abstract

Let $\mathcal{P} = \{p_1, p_2, \ldots p_n\}$ and $\mathcal{Q} = \{q_1, q_2 \ldots q_m\}$ be two point sets in an arbitrary metric space. Let $\mathbf{A}$ represent the $m \times n$ pairwise distance matrix with $\mathbf{A}_{i,j} = d(p_i, q_j)$. Such distance matrices are commonly computed in software packages and have applications to learning image manifolds, handwriting recognition, and multi-dimensional unfolding, among other things. In an attempt to reduce their description size, we study low rank approximation of such matrices. Our main result is to show that for any underlying distance metric $d$, it is possible to achieve an additive error low rank approximation in sublinear time. We note that it is provably impossible to achieve such a guarantee in sublinear time for arbitrary matrices $\mathbf{A}$, and our proof exploits special properties of distance matrices. We develop a recursive algorithm based on additive projection-cost preserving sampling. We then show that in general, relative error approximation in sublinear time is impossible for distance matrices, even if one allows for bicriteria solutions. Additionally, we show that if $\mathcal{P} = \mathcal{Q}$ and $d$ is the squared Euclidean distance, which is not a metric but rather the square of a metric, then a relative error bicriteria solution can be found in sublinear time. Finally, we empirically compare our algorithm with the singular value decomposition (SVD) and input sparsity time algorithms. Our algorithm is several hundred times faster than the SVD, and about 8-20 times faster than input sparsity methods on real-world and and synthetic datasets of size $10^8$. Accuracy-wise, our algorithm is only slightly worse than that of the SVD (optimal) and input-sparsity time algorithms.

## 1 Introduction

We study low rank approximation of matrices $\mathbf{A}$ formed by the pairwise distances between two (possibly equal) sets of points or observations $\mathcal{P} = \{p_1, \ldots, p_m\}$ and $\mathcal{Q} = \{q_1, \ldots, q_n\}$ in an arbitrary underlying metric space. That is, $\mathbf{A}$ is an $m \times n$ matrix for which $A_{i,j} = d(p_i, q_j)$. Such distance matrices are the outputs of routines in commonly used software packages such as the pairwise command in Julia, the pdist2 command in Matlab, or the crossdist command in R.

Distance matrices have found many applications in machine learning, where Weinberger and Sauk use them to learn image manifolds [18], Tenenbaum, De Silva, and Langford use them for image understanding and handwriting recognition [17], Jain and Saul use them for speech and music [12],

and Demaine et al. use them for music and musical rhythms [7]. For an excellent tutorial on Euclidean distance matrices, we refer the reader to [8], which lists applications to nuclear magnetic resonance (NMR), crystallagraphy, visualizing protein structure, and multi-dimensional unfolding.

We consider the most general case for which $\mathcal{P}$ and $\mathcal{Q}$ are not necessarily the same set of points. For example, one may have two large unordered sets of samples from some distribution, and may want to determine how similar (or dissimilar) the sample sets are to each other. Such problems arise in hierarchical clustering and phylogenetic analysis[1]. Formally, Let $\mathcal{P} = \{p_1, p_2, \ldots p_m\}$ and $\mathcal{Q} = \{q_1, q_2 \ldots q_n\}$ be two sets of points in an arbitrary metric space. Let $\mathbf{A}$ represent the $m \times n$ pairwise distance matrix with $A_{i,j} = d(p_i, q_j)$. Since the matrix $\mathbf{A}$ may be very large, it is often desirable to reduce the number of parameters needed to describe it. Two standard methods of doing this are via sparsity and low-rank approximation. In the distance matrix setting, if one first filters $\mathcal{P}$ and $\mathcal{Q}$ to contain only distinct points, then each row and column can contain at most a single zero entry, so typically such matrices $\mathbf{A}$ are dense. Low-rank approximation, on the other hand, can be highly beneficial since if the point sets can be clustered into a small number of clusters, then each cluster can be used to define an approximately rank-1 component, and so $\mathbf{A}$ is an approximately low rank matrix.

To find a low rank factorization of $\mathbf{A}$, one can compute its singular value decomposition (SVD), though in practice this takes $\min(mn^2, m^2n)$ time. One can do slightly better with theoretical algorithms for fast matrix multiplication, though not only are they impractical, but there exist much faster randomized approximation algorithms. Indeed, one can use Fast Johnson Lindenstrauss transforms (FJLT) [16], or CountSketch matrices [4, 13, 15, 1, 5], which for dense matrices $\mathbf{A}$, run in $O(mn) + (m + n)\text{poly}(k/\epsilon)$ time. At first glance the $O(mn)$ time seems like it could be optimal. Indeed, for arbitrary $m \times n$ matrices $\mathbf{A}$, outputting a rank-$k$ matrix $\mathbf{B}$ for which

$$\|\mathbf{A} - \mathbf{B}\|_F^2 \leq \|\mathbf{A} - \mathbf{A}_k\|_F^2 + \epsilon\|\mathbf{A}\|_F^2 \tag{1.1}$$

can be shown to require $\Omega(mn)$ time. Here $\mathbf{A}_k$ denotes the best rank-$k$ approximation to $\mathbf{A}$ in Frobenius norm, and recall for an $m \times n$ matrix $\mathbf{C}$, $\|\mathbf{C}\|_F^2 = \sum_{i=1,\ldots,m,j=1,\ldots,n} C_{i,j}^2$. The *additive error* guarantee above is common in low-rank approximation literature and appears in [10]. To see this lower bound, note that if one does not read nearly all the entries of $\mathbf{A}$, then with good probability one may miss an entry of $\mathbf{A}$ which is arbitrarily large, and therefore cannot achieve (1.1).

Perhaps surprisingly, [14] show that for positive semidefinite (PSD) $n \times n$ matrices $\mathbf{A}$, one can achieve (1.1) in *sublinear* time, namely, in $n \cdot k \cdot \text{poly}(1/\epsilon)$ time. Moreover, they achieve the stronger notion of relative error, that is, they output a rank-$k$ matrix $\mathbf{B}$ for which

$$\|\mathbf{A} - \mathbf{B}\|_F^2 \leq (1 + \epsilon)\|\mathbf{A} - \mathbf{A}_k\|_F^2. \tag{1.2}$$

The intuition behind their result is that the "large entries" causing the $\Omega(mn)$ lower bound cannot hide in a PSD matrix, since they necessarily create large diagonal entries. A natural question is whether it is possible to obtain low-rank approximation algorithms for distance matrices in sublinear time as well. A driving intuition that it may be possible is that no matter which metric the underlying points reside in, they necessarily satisfy the triangle inequality. Therefore, if $\mathbf{A}_{i,j} = d(p_i, q_j)$ is large, then since $d(p_i, q_j) \leq d(p_i, q_1) + d(q_1, p_1) + d(p_1, q_j)$, at least one of $d(p_i, q_1), d(q_1, p_1), d(p_1, q_j)$ is large, and further, all these distances can be found by reading the first row and column of $\mathbf{A}$. Thus, large entries cannot hide in the matrix. Are there sublinear time algorithms achieving (1.1)? Are there sublinear time algorithms achieving (1.2)? These are the questions we put forth and study in this paper.

## 1.1 Our Results

Our main result is that we obtain sublinear time algorithms achieving the additive error guarantee similar to (1.1) for distance matrices, which is impossible for general matrices $\mathbf{A}$. We show that for *every metric* $d$, this is indeed possible. Namely, for an arbitrarily small constant $\gamma > 0$, we give an algorithm running in $\widetilde{O}((m^{1+\gamma} + n^{1+\gamma})\text{poly}(k\epsilon^{-1}))$ time and achieving guarantee $\|\mathbf{A} - \mathbf{MN}^T\|_F^2 \leq \|\mathbf{A} - \mathbf{A}_k\|_F^2 + \epsilon\|\mathbf{A}\|_F^2$, for any distance matrix with metric $d$. Note that our running time is significantly *sublinear* in the description size of $\mathbf{A}$. Indeed, thinking of the shortest path metric on an unweighted bipartite graph in which $\mathcal{P}$ corresponds to the left set of vertices, and $\mathcal{Q}$ corresponds to the right set of vertices, for each pair of points $p_i \in \mathcal{P}$ and $q_j \in \mathcal{Q}$, one can choose $d(p_i, q_j) = 1$ or $d(p_i, q_j) > 1$ independently of all other distances by deciding whether to include the edge $\{p_i, q_j\}$. Consequently,

there are at least $2^{\Omega(mn)}$ possible distance matrices $\mathbf{A}$, and since our algorithm reads $o(mn)$ entries of $\mathbf{A}$, cannot learn whether $d(p_i, q_j) = 1$ or $d(p_i, q_j) > 1$ for each $i$ and $j$. Nevertheless, it still learns enough information to compute a low rank approximation to $\mathbf{A}$.

We note that a near matching lower bound holds just to write down the output of a factorization of a rank-$k$ matrix $\mathbf{B}$ into an $m \times k$ and a $k \times n$ matrix. Thus, up to an $(m^\gamma + n^\gamma)\text{poly}(k\epsilon^{-1})$ factor, our algorithm is also optimal among those achieving the additive error guarantee of (1.1).

A natural followup question is to consider achieving relative error (1.2) in sublinear time. Although large entries in a distance matrix $\mathbf{A}$ cannot hide, we show it is still impossible to achieve the relative error guarantee in less than $mn$ time for distance matrices. That is, we show for the $\ell_\infty$ distance metric, that there are instances of distance matrices $\mathbf{A}$ with unequal $\mathcal{P}$ and $\mathcal{Q}$ for which even for $k = 2$ and any constant accuracy $\epsilon$, must read $\Omega(mn)$ entries of $\mathbf{A}$. In fact, our lower bound holds even if the algorithm is allowed to output a rank-$k'$ approximation for any $2 \le k' = o(\min(m, n))$ whose cost is at most that of the best rank-2 approximation to $\mathbf{A}$. We call the latter a bicriteria algorithm, since its output rank $k'$ may be larger than the desired rank $k$. Therefore, in some sense obtaining additive error (1.1) is the best we can hope for.

We next consider the important class of Euclidean matrices for which the entries correspond to the *square* of the Euclidean distance, and for which $\mathcal{P} = \mathcal{Q}$. In this case, we are able to show that if we allow the low rank matrix $\mathbf{B}$ output to be of rank $k + 4$, then one can achieve the relative error guarantee of (1.2) with respect to the best rank-$k$ approximation, namely, that

$$\|\mathbf{A} - \mathbf{B}\|_F^2 \le (1 + \epsilon)\|\mathbf{A} - \mathbf{A}_k\|_F^2.$$

Further, our algorithm runs in a sublinear $n \cdot k \cdot \text{poly}(1/\epsilon)$ amount of time. Thus, our lower bound ruling out sublinear time algorithms achieving (1.2) for bicriteria algorithms cannot hold for this class of matrices.

Finally, we empirically compare our algorithm with the SVD and input sparsity time algorithms [4, 13, 15, 1, 5]. Our algorithm is several hundred times faster than the SVD, and about 8-20 times faster than input sparsity methods on real-world datasets such as MNIST, Gisette and Poker, and a synthetic clustering dataset of size $10^8$. Accuracy-wise, our algorithm is only slightly worse than that of the SVD (optimal error) and input-sparsity time algorithms. Due to space constraints, we defer all of our proofs to the Supplementary Material [2].

## 2 Row and Column Norm Estimation

We observe that we can obtain a rough estimate for the row or column norms of a distance matrix by uniformly sampling a small number of elements of each row or column. The only structural property we need to obtain such an estimate is approximate triangle inequality.

**Definition 2.1.** *(Approximate Triangle Inequality.) Let $\mathbf{A}$ be a $m \times n$ matrix. Then, matrix $\mathbf{A}$ satisfies approximate triangle inequality if, for any $\epsilon \in [0, 1]$, for any $p \in [m]$, $q, r \in [n]$*

$$\frac{|\mathbf{A}_{p,r} - \max_{i \in [m]} |\mathbf{A}_{i,q} - \mathbf{A}_{i,r}||}{(1 + \epsilon)} \le \mathbf{A}_{p,q} \le (1 + \epsilon)\left(\mathbf{A}_{p,r} + \max_{i \in [m]} |\mathbf{A}_{i,q} - \mathbf{A}_{i,r}|\right) \quad (2.1)$$

$$\frac{|\mathbf{A}_{p,q} - \mathbf{A}_{p,r}|}{1 + \epsilon} \le \max_{i \in [m]} |\mathbf{A}_{i,q} - \mathbf{A}_{i,r}| \le (1 + \epsilon)\left(\mathbf{A}_{p,q} + \mathbf{A}_{p,r}\right) \quad (2.2)$$

*Further, similar equations hold for $\mathbf{A}^T$.*

The above definition captures distance matrices if we set $\epsilon = 0$. In order to see this, recall, each entry in a $m \times n$ matrix $\mathbf{A}$ is associated with a distance between points sets $\mathcal{P}$ and $\mathcal{Q}$, such that $|\mathcal{P}| = m$ and $|\mathcal{Q}| = n$. Then, for points $p \in \mathcal{P}$, $q \in \mathcal{Q}$, $\mathbf{A}_{p,q}$ represents $d(p, q)$, where $d$ is an arbitrary distance metric. Further, for arbitrary point, $i \in \mathcal{P}$ and $r \in \mathcal{Q}$, $\max_i |\mathbf{A}_{i,q} - \mathbf{A}_{i,r}| = \max_i |d(i, r) - d(i, q)|$. Intuitively, we would like to highlight that, for the case where $\mathbf{A}$ is a distance matrix, $\max_{i \in [m]} |\mathbf{A}_{i,q} - \mathbf{A}_{i,r}|$ represents a lower bound on the distance $d(q, r)$. Since $d$ is a metric, it follows triangle inequality, and $d(p, q) \le d(p, r) + d(q, r)$. Further, by reverse triangle inequality, for all $i \in [m]$, $d(q, r) \ge |d(i, q) - d(i, r)|$. Therefore, $\mathbf{A}_{p,q} \le \mathbf{A}_{p,r} + \max_i |\mathbf{A}_{i,q} - \mathbf{A}_{i,r}|$ and distance matrices satisfy equation 2.1. Next, $\max_{i \in [m]} |\mathbf{A}_{i,q} - \mathbf{A}_{i,r}| = \max_{i \in [m]} |d_{i,q} - d_{i,r}| \le d(q, r)$,

and $d(q,r) \leq d(p,r) + d(p,q) = \mathbf{A}_{p,r} + \mathbf{A}_{p,q}$ therefore, equation 2.2 is satisfied. We note that approximate triangle inequality is a relaxation of the traditional triangle inequality and is sufficient to obtain coarse estimates to row and column norms of $\mathbf{A}$ in sublinear time.

---

**Algorithm 1 : Row Norm Estimation.**

---

**Input:** A Distance Matrix $\mathbf{A}_{m \times n}$, Sampling parameter $b$.

1. Let $x = \text{argmin}_{i \in [m]} \mathbf{A}_{i,1}$.

2. Let $d = \max_{j \in [n]} \mathbf{A}_{x,j}$.

3. For $i \in [m]$, let $\mathcal{T}_i$ be a uniformly random sample of $\Theta(b)$ indices in $[n]$.

4. $\widetilde{X}_i = d^2 + \sum_{j \in \mathcal{T}_i} \frac{n}{b} \mathbf{A}_{i,j}^2$.

**Output:** Set $\{\widetilde{X}_1, \widetilde{X}_2, \ldots \widetilde{X}_m\}$

---

**Lemma 2.1.** *(Row Norm Estimation.) Let $\mathbf{A}$ be a $m \times n$ matrix such that $\mathbf{A}$ satisfies approximate triangle inequality. For $i \in [m]$, let $\mathbf{A}_{i,*}$ be the $i^{th}$ row of $\mathbf{A}$. Algorithm 1 uniformly samples $\Theta(b)$ elements from $\mathbf{A}_{i,*}$ and with probability at least $9/10$ outputs an estimator which obtains an $O(n/b)$-approximation to $\|\mathbf{A}_{i,*}\|_2^2$. Further, Algorithm 1 runs in $O(bm + n)$ time.*

To obtain an $O(n/b)$ approximation for all the $m$ rows simultaneously with high probability, we can compute $O(\log(m))$ estimators for each row and take their median. We also observe that Column and Row Norm Estimation are symmetric operations and a slight modification to Algorithm 1 yields a Column Norm Estimation algorithm with the following guarantee:

**Corollary 2.1.** *(Column Norm Estimation.) There exists an algorithm that uniformly samples $\Theta(b)$ elements from $\mathbf{A}_{*,j}$ and with probability $9/10$, outputs an estimator which is an $O(m/b)$-approximation to $\|\mathbf{A}_{*,j}\|_2^2$. Further, this algorithm runs in $O(bn + m)$ time.*

# 3 Projection-Cost Preserving Sketches

We would like to reduce the dimensionality of the matrix in a way that approximately preserves low-rank structure. The main insight is that if we can approximately preserve all rank-$k$ subspaces in the column and row space of the matrix, then we can recursively sample rows and columns to obtain a much smaller matrix. To this end, we introduce a relaxation of projection-cost preserving sketches [6] that satisfy an *additive error guarantee*. We show that sampling columns of $\mathbf{A}$ according to approximate column norms yields a matrix that preserves projection cost for all rank-$k$ projections up to additive error. We give a similar proof to the relative error guarantees in [6], but need to replace certain parts with our different distribution which is only based on row and column norms rather than leverage scores, and consequently we obtain additive error in places instead. As our lower bound shows, this is necessary in our setting.

**Theorem 3.1.** *(Column Projection-Cost Preservation.) Let $\mathbf{A}$ be a $m \times n$ matrix such that $\mathbf{A}$ satisfies approximate triangle inequality. For $j \in [n]$, let $\widetilde{X}_j$ be an $O(m/b)$-approximate estimate for the $j^{th}$ column of $\mathbf{A}$ such that it satisfies the guarantee of Corollary A.1. Then, let $q = \{q_1, q_2 \ldots q_n\}$ be a probability distribution over the columns of $\mathbf{A}$ such that $q_j = \widetilde{X}_j / \sum_{j'} \widetilde{X}_{j'}$. Let $t = O\left(\frac{mk^2}{b\epsilon^2} \log(\frac{m}{\delta})\right)$ for some constant c. Then, construct $\mathbf{C}$ using $t$ columns of $\mathbf{A}$ and set each one to $\mathbf{A}_{*,j}/\sqrt{tq_j}$ with probability $q_j$. With probability at least $1 - \delta$, for any rank-$k$ orthogonal projection $\mathbf{X}$, $\|\mathbf{C} - \mathbf{XC}\|_F^2 = \|\mathbf{A} - \mathbf{XA}\|_F^2 \pm \epsilon\|\mathbf{A}\|_F^2$.*

We observe that we can also estimate the row norms in sublinear time and immediately obtain a similar guarantee for *row projection-cost preservation*. Next, we describe how to apply projection-cost preserving sketching for low-rank approximation. Let $\mathbf{C}$ be a *column pcp* for $\mathbf{A}$. Then, an approximate solution for the best rank-$k$ approximation to $\mathbf{C}$ is an approximate solution for the best rank-$k$ approximation to $\mathbf{A}$. Formally,

**Lemma 3.1.** *Let $\mathbf{C}$ be a column pcp for $\mathbf{A}$ satisfying the guarantee of Theorem A.2. Let $\mathbf{P}_{\mathbf{C}}^*$ be the minimizing projection matrix for $\min_{\mathbf{X}} \|\mathbf{C} - \mathbf{XC}\|_F^2$ and $\mathbf{P}_{\mathbf{A}}^*$ be the projection matrix that minimizes $\min_{\mathbf{X}} \|\mathbf{A} - \mathbf{XA}\|_F^2$. Then, for any projection matrix $\mathbf{P}$ such that $\|\mathbf{C} - \mathbf{PC}\|_F^2 \leq \|\mathbf{C} - \mathbf{P}_{\mathbf{C}}^*\mathbf{C}\|_F^2 + \epsilon\|\mathbf{C}\|_F^2$, with probability at least $98/100$, $\|\mathbf{A} - \mathbf{PA}\|_F^2 \leq \|\mathbf{A} - \mathbf{P}_{\mathbf{A}}^*\mathbf{A}\|_F^2 + \epsilon\|\mathbf{A}\|_F^2$. A similar guarantee holds if $\mathbf{C}$ is a row pcp of $\mathbf{A}$.*

# 4 A Sublinear Time Algorithm.

---

**Algorithm 2 : First Sublinear Time Algorithm.**

---

**Input:** A Distance Matrix $\mathbf{A}_{m \times n}$, integer $k$ and $\epsilon > 0$.

1. Set $b_1 = \frac{\epsilon n^{0.34}}{\log(n)}$ and $b_2 = \frac{\epsilon m^{0.34}}{\log(m)}$. Set $s_1 = \Theta\left(\frac{mk^2 \log(m)}{b_1 \epsilon^2}\right)$ and $s_2 = \Theta\left(\frac{nk^2 \log(n)}{b_2 \epsilon^2}\right)$.

2. Let $\widetilde{X}_j$ be the estimate for $\|\mathbf{A}_{*,j}\|_2^2$ returned by ColumnNormEstimation$(\mathbf{A}, b_1)$. Recall, $\widetilde{X}_j$ is an $O\left(\frac{m}{b_1}\right)$-approximation to $\mathbf{A}_{*,j}$.

3. Let $q = \{q_1, q_2 \ldots q_n\}$ denote a distribution over columns of $\mathbf{A}$ such that $q_i = \frac{\widetilde{X}_j}{\sum_j \widetilde{X}_j} \geq \left(\frac{b_1}{m}\right)\frac{\|\mathbf{A}_{*,j}\|_2^2}{\|\mathbf{A}\|_F^2}$. Construct a *column pcp* for $\mathbf{A}$ by sampling $s_1$ columns of $\mathbf{A}$ such that each column is set to $\frac{\mathbf{A}_{*,j}}{\sqrt{s_1 q_j}}$ with probability $q_j$. Let $\mathbf{AS}$ be the resulting $m \times s_1$ matrix that follows guarantees of Theorem A.2.

4. To account for the rescaling, consider $O(\epsilon^{-1} \log(n))$ weight classes for scaling parameters of the columns of $\mathbf{AS}$. Let $\mathbf{AS}_{|\mathcal{W}_g}$ be the columns of $\mathbf{AS}$ restricted to the weight class $\mathcal{W}_g$ (defined below.)

5. Run the RowNormEstimation$(\mathbf{AS}_{|\mathcal{W}_g}, b_2)$ estimation algorithm with parameter $b_2$ for each weight class independently and sum up the estimates for a given row. Let $\widetilde{X}_i$ be the resulting $O\left(\frac{n}{b_2}\right)$-approximate estimator for $\mathbf{AS}_{i,*}$.

6. Let $p = \{p_1, p_2, \ldots p_m\}$ denote a distribution over rows of $\mathbf{AS}$ such that $p_i = \frac{\widetilde{X}_i}{\sum_i \widetilde{X}_i} \geq \left(\frac{b_2}{n}\right)\frac{\|\mathbf{AS}_{i,*}\|_2^2}{\|\mathbf{AS}\|_F^2}$. Construct a *row pcp* for $\mathbf{AS}$ by sampling $s_2$ rows of $\mathbf{AS}$ such that each row is set to $\frac{\mathbf{AS}_{i,*}}{\sqrt{s_2 p_i}}$ with probability $p_i$. Let $\mathbf{TAS}$ be the resulting $s_2 \times s_1$ matrix that follows guarantees of Corollary A.2.

7. Run the input-sparsity time low-rank approximation algorithm (corresponding to Theorem 4.2) on $\mathbf{TAS}$ with rank parameter $k$ to obtain a rank-$k$ approximation to $\mathbf{TAS}$, output in factored form: $\mathbf{L}, \mathbf{D}, \mathbf{W}^T$. Note, $\mathbf{LD}$ is an $s_2 \times k$ matrix and $\mathbf{W}^T$ is a $k \times s_1$ matrix.

8. Consider the regression problem $\min_{\mathbf{X}} \|\mathbf{AS} - \mathbf{XW}^T\|_F^2$. Sketch the problem using the leverage scores of $\mathbf{W}^T$ as shown in Theorem 4.3 to obtain a sampling matrix $\mathbf{E}$ with poly$(\frac{k}{\epsilon})$ columns. Compute $\mathbf{X_{AS}} = \text{argmin}_{\mathbf{X}} \|\mathbf{ASE} - \mathbf{XW}^T\mathbf{E}\|_F^2$. Let $\mathbf{X_{AS}}\mathbf{W}^T = \mathbf{P}'\mathbf{N}'^T$ be such that $\mathbf{P}'$ has orthonormal columns.

9. Consider the regression problem $\min_{\mathbf{X}} \|\mathbf{A} - \mathbf{P}'\mathbf{X}\|_F^2$. Sketch the problem using the the leverage scores of $\mathbf{P}'$ following Theorem 4.3 to obtain a sampling matrix $\mathbf{E}'$ with poly$(\frac{k}{\epsilon})$ rows. Compute $\mathbf{X_A} = \text{argmin}_{\mathbf{X}} \|\mathbf{E}'\mathbf{A} - \mathbf{E}'\mathbf{P}'\mathbf{X}\|_F^2$.

**Output:** $\mathbf{M} = \mathbf{P}'$, $\mathbf{N}^T = \mathbf{X_A}$

---

In this section, we give a sublinear time algorithm which relies on constructing *column and row pcps*, which in turn rely on our column and row norm estimators. Intuitively, we begin with obtaining coarse estimates to column norms. Next, we sample a subset of the columns of $\mathbf{A}$ with probability proportional to their column norm estimates to obtain a *column pcp* for $\mathbf{A}$. We show that the rescaled matrix still has enough structure to get a coarse estimate to its row norms. Then, we compute the row norm estimates of the sampled rescaled matrix, and subsample its rows to obtain a small matrix that is a *row pcp*. We run an input-sparsity time algorithm ([4]) on the small matrix to obtain a low-rank approximation. The main theorem we prove is as follows:

**Theorem 4.1.** *(Sublinear Low-Rank Approximation.) Let $\mathbf{A} \in \mathbb{R}^{m \times n}$ be a matrix that satisfies approximate triangle inequality. Then, for any $\epsilon > 0$ and integer $k$, Algorithm 2 runs in time $O\left((m^{1.34} + n^{1.34}) \text{poly}(\frac{k}{\epsilon})\right)$ and outputs matrices $\mathbf{M} \in \mathbb{R}^{m \times k}$, $\mathbf{N} \in \mathbb{R}^{n \times k}$ such that with probability at least $9/10$,*

$$\left\|\mathbf{A} - \mathbf{MN}^T\right\|_F^2 \leq \|\mathbf{A} - \mathbf{A}_k\|_F^2 + \epsilon \|\mathbf{A}\|_F^2$$

**Column Sampling.** We observe that constructing a column and row projection-cost preserving sketches require sampling columns proportional to their relative norms and subsequently subsample columns proportional to the relative row norms of the sampled, rescaled matrix. In the previous section, we obtain coarse approximations to these norms and thus use our estimates to serve as a proxy for the real distribution. For $j \in [n]$, let $\widetilde{X}_j$ be our estimate for the column $\mathbf{A}_{*,j}$. We define a probability distribution over the columns of $\mathbf{A}$ as $q_j = \frac{\widetilde{X}_j}{\sum_{i' \in [n]} \widetilde{X}_{j'}}$. Given that we can estimate column norms up to an $O(m/b_1)$-factor, $q_j \geq \left(\frac{b_1}{m}\right) \frac{\|\mathbf{A}_{*,j}\|_2^2}{\|\mathbf{A}\|_F^2}$, where $b_1$ is a parameter to be set later. Therefore, we oversample columns of $\mathbf{A}$ by a $\Theta(m/b_1)$-factor to construct a *column pcp* for $\mathbf{A}$. Let $\mathbf{AS}$ be a scaled sample of $s_1 = \Theta\left(\frac{mk^2 \log(m)}{b_1 \epsilon^2}\right)$ columns of $\mathbf{A}$ such that each column is set to $\frac{\mathbf{A}_{*,j}}{\sqrt{s_1 q_j}}$ with probability $q_j$. Then, by Theorem A.2 for any $\epsilon > 0$, with probability at least $99/100$, for all rank-$k$ projection matrices $\mathbf{X}$, $\|\mathbf{AS} - \mathbf{XAS}\|_F^2 \leq \|\mathbf{A} - \mathbf{XA}_k\|_F^2 + \epsilon \|\mathbf{A}\|_F^2$.

**Handling Rescaled Columns.** We note that during the construction of the *column pcp*, the $j^{th}$ column, if sampled, is rescaled by $\frac{1}{\sqrt{s_1 q_j}}$. Therefore, the resulting matrix, $\mathbf{AS}$ may no longer be a distance matrix. To address this issue, we partition the columns of $\mathbf{AS}$ into weight classes such that the $g^{th}$ weight class contains column index $j$ if the corresponding scaling factor $\frac{1}{\sqrt{q_j}}$ lies in the interval $[(1+\epsilon)^g, (1+\epsilon)^{g+1})$. Note, we can ignore the $(\frac{1}{\sqrt{s_1}})$-factor since every entry is rescaled by the same constant. Formally, $\mathcal{W}_g = \left\{ i \in [s_1] \,\middle|\, \frac{1}{\sqrt{q_j}} \in [(1+\epsilon)^g, (1+\epsilon)^{g+1}) \right\}$. Next, with high probability, for all $j \in [n]$, if column $j$ is sampled, $1/q_j \leq n^c$ for a large constant $c$. If instead, $q_j \leq \frac{1}{n^{c'}}$, the probability that the $j^{th}$ is sampled would be at most $1/n^{c'}$, for some $c' > c$. Union bounding over such events for $n$ columns, the number of weight classes is at most $\log_{1+\epsilon}(n^c) = O\left(\epsilon^{-1} \log(n)\right)$. Let $\mathbf{AS}_{|\mathcal{W}_g}$ denote the columns of $\mathbf{AS}$ restricted to the set of indices in $\mathcal{W}_g$. Observe that all entries in $\mathbf{AS}_{|\mathcal{W}_g}$ are scaled to within a $(1+\epsilon)$-factor of each other and therefore, satisfy approximate triangle inequality (equation 2.1).

Therefore, row norms of $\mathbf{AS}_{|\mathcal{W}_g}$ can be computed using Algorithm 1 and the estimator is an $O(n/b_2)$-approximation (for some parameter $b_2$), since Lemma 2.1 blows up by a factor of at most $(1+\epsilon)$. Summing over the estimates from each partition above, with probability at least $99/100$, we obtain an $O(n/b_2)$-approximate estimate to $\|\mathbf{AS}_{i,*}\|_2^2$, simultaneously for all $i \in [m]$. However, we note that each iteration of Algorithm 1 reads $b_2 m + n$ entries of $\mathbf{A}$ and there are at most $O(\epsilon^{-1} \log(n))$ iterations.

**Row Sampling.** Next, we construct a *row pcp* for $\mathbf{AS}$. For $i \in [m]$, let $\widetilde{X}_i$ be an $O(n/b_2)$-approximate estimate for $\|\mathbf{AS}_{i,*}\|_2^2$. Let $p = \{p_1, p_2, \ldots, p_m\}$ be a distribution over the rows of $\mathbf{AS}$ such that $p_i = \frac{\widetilde{X}_i}{\sum_i \widetilde{X}_i} \geq \left(\frac{b_2}{n} \frac{\|\mathbf{AS}_{i,*}\|_2^2}{\|\mathbf{AS}\|_F^2}\right)$. Therefore, we oversample rows by a $\Theta(n/b_2)$ factor to obtain a *row pcp* for $\mathbf{AS}$. Let $\mathbf{TAS}$ be a scaled sample of $s_2 = \Theta\left(\frac{nk^2 \log(n)}{b_2 \epsilon^2}\right)$ rows of $\mathbf{AS}$ such that each row is set to $\frac{\mathbf{AS}_{i,*}}{\sqrt{s_2 p_i}}$ with probability $p_i$. By the row analogue of Theorem A.2, with probability at least $99/100$, for all rank-$k$ projection matrices $\mathbf{X}$, $\|\mathbf{TAS} - \mathbf{TASX}\|_F^2 \leq \|\mathbf{AS} - \mathbf{ASX}\|_F^2 + \epsilon \|\mathbf{AS}\|_F^2$.

**Input-sparsity Time Low-Rank Approximation.** Next, we compute a low-rank approximation for the smaller matrix, $\mathbf{TAS}$, in input-sparsity time. To this end we use the following theorem from [4]:

**Theorem 4.2.** *(Clarkson-Woodruff LRA.) For $\mathbf{A} \in \mathbb{R}^{m \times n}$, there is an algorithm that with failure probability at most $1/10$ finds $L \in R^{m \times k}$, $W \in R^{n \times k}$ and a diagonal matrix $D \in R^{k \times k}$, such that $\|\mathbf{A} - \mathbf{LDW}^T\|_F^2 \leq (1+\epsilon) \|\mathbf{A} - \mathbf{A}_k\|_F^2$ and runs in time $O\left(\mathbf{nnz}(\mathbf{A}) + (n+m)\mathrm{poly}(\frac{k}{\epsilon})\right)$, where $\mathbf{nnz}(\mathbf{A})$ is the number of non-zero entries in $\mathbf{A}$.*

Running the input-sparsity time algorithm with the above guarantee on the matrix $\mathbf{TAS}$, we obtain a rank-$k$ matrix $\mathbf{LDW}^T$, such that $\|\mathbf{TAS} - \mathbf{LDW}^T\|_F^2 \leq (1+\epsilon)\|\mathbf{TAS} - (\mathbf{TAS})_k\|_F^2$ where $(\mathbf{TAS})_k$ is the best rank-$k$ approximation to $\mathbf{TAS}$ under the Frobenius norm. Since $\mathbf{TAS}$ is a small matrix, we can afford to read all of it by querying at most $O\left(\frac{nm \log(n) \log(m)}{b_1 b_2} \mathrm{poly}(\frac{k}{\epsilon})\right)$ entries of $\mathbf{A}$ and the algorithm runs in time $O\left(s_1 s_2 + (s_1 + s_2)\mathrm{poly}(\frac{k}{\epsilon})\right)$.

**Constructing a solution for $\mathbf{A}$.** Note, while $\mathbf{LDW}^T$ is an approximate rank-$k$ solution for $\mathbf{TAS}$, it does not have the right dimensions as $\mathbf{A}$. If we do not consider running time, we could construct a low-rank approximation to $\mathbf{A}$ as follows: since projecting $\mathbf{TAS}$ onto $\mathbf{W}^T$ is approximately optimal, it follows from Lemma A.1 that with probability $98/100$,

$$\|\mathbf{AS} - \mathbf{ASWW}^T\|_F^2 = \|\mathbf{AS} - (\mathbf{AS})_k\|_F^2 \pm \epsilon \|\mathbf{AS}\|_F^2 \tag{4.1}$$

Let $(\mathbf{AS})_k = \mathbf{PN}^T$ be such that $\mathbf{P}$ has orthonormal columns. Then, $\|\mathbf{AS} - \mathbf{PP}^T\mathbf{AS}\|_F^2 = \|\mathbf{AS} - (\mathbf{AS})_k\|_F^2$ and by Lemma A.1 it follows that with probability $98/100$, $\|\mathbf{A} - \mathbf{PP}^T\mathbf{A}\|_F^2 \leq \|\mathbf{A} - \mathbf{A}_k\|_F^2 + \epsilon \|\mathbf{A}\|_F$. However, even approximately computing a column space $\mathbf{P}$ for $(\mathbf{AS})_k$ using an input-sparsity time algorithm is no longer sublinear. To get around this issue, we observe that an approximate solution for $\mathbf{TAS}$ lies in the row space of $\mathbf{W}^T$ and therefore, an approximately optimal solution for $\mathbf{AS}$ lies in the row space of $\mathbf{W}^T$. We then set up the following regression problem: $\min_{\mathbf{X}} \|\mathbf{AS} - \mathbf{XW}^T\|_F^2$.

Note, this regression problem is still too big to be solved in sublinear time. Therefore, we sketch it by sampling columns of $\mathbf{AS}$ according to the leverage scores of $\mathbf{W}^T$ to set up a smaller regression problem. Formally, we use a theorem of [4] (Theorem 38) to approximately solve this regression problem (also see [9] for previous work.)

**Theorem 4.3.** *(Fast Regression.) Given a matrix $\mathbf{A} \in \mathbb{R}^{m \times n}$ and a rank-$k$ matrix $\mathbf{B} \in \mathbb{R}^{m \times k}$, such that $\mathbf{B}$ has orthonormal columns, the regression problem $\min_{\mathbf{X}} \|\mathbf{A} - \mathbf{BX}\|_F^2$ can be solved up to $(1 + \epsilon)$ relative error, with probability at least $2/3$ in time $O\left((m \log(m) + n)poly(\frac{k}{\epsilon})\right)$ by constructing a sketch $\mathbf{E}$ with $poly(\frac{k}{\epsilon})$ rows and solving $\min_{\mathbf{X}} \|\mathbf{EA} - \mathbf{EBX}\|_F^2$. Note, a similar guarantee holds for solving $\min_{\mathbf{X}} \|\mathbf{A} - \mathbf{XB}\|_F^2$.*

Since $\mathbf{W}^T$ has orthonomal rows, the leverage scores are precomputed. With probability at least $99/100$, we can compute $\mathbf{X_{AS}} = \text{argmin}_{\mathbf{X}} \|\mathbf{ASE} - \mathbf{XW}^T\mathbf{E}\|_F^2$, where $\mathbf{E}$ is a leverage score sketching matrix with $poly(\frac{k}{\epsilon})$ columns, as shown in Theorem 4.3.

$$\|\mathbf{AS} - \mathbf{X_{AS}}\mathbf{W}^T\|_F^2 \leq (1+\epsilon)\min_{\mathbf{X}}\|\mathbf{AS} - \mathbf{XW}^T\|_F^2 \leq (1+\epsilon)\|\mathbf{AS} - \mathbf{ASWW}^T\|_F^2$$
$$= \|\mathbf{AS} - (\mathbf{AS})_k\|_F^2 \pm \epsilon\|\mathbf{AS}\|_F^2 \tag{4.2}$$

where the last two inequalities follow from equation 4.1. Recall, $\mathbf{AS}$ is an $m \times s_1$ matrix and thus the running time is $O\left((m + s_1)\log(m)poly\left(\frac{k}{\epsilon}\right)\right)$. Let $\mathbf{X_{AS}}\mathbf{W}^T = \mathbf{P}'\mathbf{N}'^T$ be such that $\mathbf{P}'$ has orthonormal columns. Then, the column space of $\mathbf{P}'$ contains an approximately optimal solution for $\mathbf{A}$, since $\|\mathbf{AS} - \mathbf{P}'\mathbf{N}'^T\|_F^2 = \|\mathbf{AS} - (\mathbf{AS})_k\|_F^2 \pm \epsilon\|\mathbf{AS}\|_F^2$ and $\mathbf{AS}$ is a *column pcp* for $\mathbf{A}$. It follows from Lemma A.1 that with probability at least $98/100$,

$$\|\mathbf{A} - \mathbf{P}'\mathbf{P}'^T\mathbf{A}\|_F^2 \leq \|\mathbf{A} - \mathbf{A}_k\|_F + \epsilon\|\mathbf{A}\|_F \tag{4.3}$$

Therefore, there exists a good solution for $\mathbf{A}$ in the column space of $\mathbf{P}'$. Since we cannot compute this explicitly, we set up the following regression problem: $\min_{\mathbf{X}} \|\mathbf{A} - \mathbf{P}'\mathbf{X}\|_F^2$. Again, we sketch the regression problem above by sampling columns of $\mathbf{A}$ according to the leverage scores of $\mathbf{P}'$. We can then compute $\mathbf{X_A} = \text{argmin}_{\mathbf{X}} \|\mathbf{E}'\mathbf{A} - \mathbf{E}'\mathbf{P}'\mathbf{X}\|_F^2$ with probability at least $99/100$, where $\mathbf{E}'$ is a leverage score sketching matrix with $poly\left(\frac{k}{\epsilon}\right)$ rows. Then, using the properties of leverage score sampling from Theorem 4.3,

$$\|\mathbf{A} - \mathbf{P}'\mathbf{X_A}\|_F^2 \leq (1+\epsilon)\min_{\mathbf{X}}\|\mathbf{A} - \mathbf{P}'\mathbf{X}\|_F^2 \leq (1+\epsilon)\|\mathbf{A} - \mathbf{P}'\mathbf{P}'^T\mathbf{A}\|_F^2$$
$$\leq \|\mathbf{A} - \mathbf{A}_k\|_F^2 + O(\epsilon)\|\mathbf{A}\|_F^2 \tag{4.4}$$

where the second inequality follows from $\mathbf{X}$ being the minimizer and $\mathbf{P}'^T\mathbf{A}$ being some other matrix, and the last inequality follows from equation 4.3. Recall, $\mathbf{P}'$ is an $m \times k$ matrix and by Theorem 38 of CW, the time taken to solve the regression problem is $O\left((m \log(m) + n)poly\left(\frac{k}{\epsilon}\right)\right)$. Therefore, we observe that $\mathbf{P}'\mathbf{X_A}$ suffices and we output it in factored form by setting $\mathbf{M} = \mathbf{P}'$ and $\mathbf{N} = \mathbf{X_A}^T$. Union bounding over the probabilistic events, and rescaling $\epsilon$, with probability at least $9/10$, Algorithm 2 outputs $\mathbf{M} \in \mathbf{R}^{m \times k}$ and $\mathbf{N} \in \mathbf{R}^{n \times k}$ such that the guarantees of Theorem 4.1 are satisfied.

Finally, we analyze the overall running time of Algorithm 2. Computing the estimates for the column norms and constructing the *column pcp* for $\mathbf{A}$ has running time $O\left(\frac{m \log(m)}{b_1}poly(\frac{k}{\epsilon}) + b_1 n + m\right)$.

| Metric | SVD | IS | Sublinear |
|--------|--------|------|-----------|
| $L_2$ | 398.76 | 8.94 | 1.69 |
| $L_1$ | 410.60 | 8.15 | 1.81 |
| $L_\infty$ | 427.90 | 9.18 | 1.63 |
| $L_c$ | 452.16 | 8.49 | 1.76 |

Table 1: Running Time (in seconds) on the Clustering Dataset for Rank = 20

| Metric | SVD | IS | Sublinear |
|--------|--------|-------|-----------|
| $L_2$ | 398.50 | 34.32 | 4.16 |
| $L_1$ | 560.9 | 39.51 | 3.72 |
| $L_\infty$ | 418.01 | 39.32 | 3.99 |
| $L_c$ | 390.07 | 38.33 | 3.91 |

Table 2: Running Time (in seconds) on the MNIST Dataset for Rank = 40

A similar guarantee holds for the rows. The input-sparsity time algorithm low-rank approximation runs in $O\left(s_1 s_2 + (s_1 + s_2)\text{poly}(\frac{k}{\epsilon})\right)$ and constructing a solution for $\mathbf{A}$ is dominated by $O\left((m \log(m) + n)\text{poly}\left(\frac{k}{\epsilon}\right)\right)$. Setting $b_1 = \frac{\epsilon n^{0.34}}{\log(n)}$ and $b_2 = \frac{\epsilon m^{0.34}}{\log(m)}$, we note that the overall running time is $\widetilde{O}\left((m^{1.34} + n^{1.34})\text{poly}\left(\frac{k}{\epsilon}\right)\right)$ where $\widetilde{O}$ hides $\log(m)$ and $\log(n)$ factors. This completes the proof of Theorem 4.1.

We extend the above result to obtain a better exponent in the running time. The critical observation here is that we can recursively sketch rows and columns of $\mathbf{A}$ such that all the rank-$k$ projections are preserved. Intuitively, it is important to preserve all rank-$k$ subspaces since we do not know which projection will be approximately optimal when we recurse back up. At a high level, the algorithm is then to recursively sub-sample columns and rows of $\mathbf{A}$ such that we obtain *projection-cost preserving sketches* at each step. We defer the details to the Supplementary Material.

**Theorem 4.4.** *Let $\mathbf{A} \in \mathbb{R}^{m \times n}$ be a matrix such that it satisfies approximate triangle inequality. Then, for any $\epsilon > 0$, integer $k$ and a small constant $\gamma > 0$, there exists an algorithm that runs in time $\widetilde{O}\left((m^{1+\gamma} + n^{1+\gamma})\,\text{poly}(\frac{k}{\epsilon})\right)$ to output matrices $\mathbf{M} \in \mathbb{R}^{m \times k}$ and $\mathbf{N} \in \mathbb{R}^{n \times k}$ such that with probability at least $9/10$, $\|\mathbf{A} - \mathbf{M}\mathbf{N}^T\|_F^2 \leq \|\mathbf{A} - \mathbf{A}_k\|_F^2 + \epsilon\|\mathbf{A}\|_F^2$.*

## 5 Relative Error Guarantees

In this section, we consider the *relative error* guarantee 1.2 for distance matrices. We begin by showing a lower bound for any relative error approximation for distance matrices and also preclude the possibility of a sublinear bi-criteria algorithm outputting a rank-poly($k$) matrix satisfying the rank-$k$ relative error guarantee.

**Theorem 5.1.** *Let $\mathbf{A}$ be an $n \times n$ distance matrix. Let $\mathbf{B}$ be a rank-poly($k$) matrix such that $\|\mathbf{A} - \mathbf{B}\|_F^2 \leq c\|\mathbf{A} - \mathbf{A}_k\|$ for any constant $c > 1$. Then, any algorithm that outputs such a $\mathbf{B}$ requires $\Omega(nnz(\mathbf{A}))$ time.*

**Euclidean Distance Matrices.** We show that in the special case of Euclidean distances, when the entries correspond to *squared* distances, there exists a bi-criteria algorithm that outputs a rank-$(k + 4)$ matrix satisfying the relative error rank-$k$ low-rank approximation guarantee. Note, here the point sets $P$ and $Q$ are identical. Let $\mathbf{A}$ be such a matrix s.t. $\mathbf{A}_{i,j} = \|x_i - x_j\|_2^2 = \|x_i\|_2^2 + \|x_j\|_2^2 - 2\langle x_i, x_j \rangle$. Then, we can write $\mathbf{A}$ as $\mathbf{A}_1 + \mathbf{A}_2 - 2\mathbf{B}$ such that each entry in the $i^{th}$ row of $\mathbf{A}_1$ is $\|x_i\|_2^2$, each entry in the $j^{th}$ column of $\mathbf{A}_2$ is $\|x_j\|_2^2$ and $\mathbf{B}$ is a PSD matrix, where $\mathbf{B}_{i,j} = \langle x_i, x_j \rangle$. The main ingredient we use is the sublinear low-rank approximation of PSD matrices from [14]. We show that there exists an algorithm (see Supplementary Material) that outputs the description of a rank-$(k + 4)$ matrix $\mathbf{A}\mathbf{W}\mathbf{W}^T$ in sublinear time such it satisfies the relative-error rank-$k$ low rank approximation guarantee.

**Theorem 5.2.** *Let $\mathbf{A}$ be a Euclidean Distance matrix. Then, for any $\epsilon > 0$ and integer $k$, there exists an algorithm that with probability at least $9/10$, outputs a rank $(k + 4)$ matrix $\mathbf{W}\mathbf{W}^T$ such that $\|\mathbf{A} - \mathbf{A}\mathbf{W}\mathbf{W}^T\|_F \leq (1 + \epsilon)\|\mathbf{A} - \mathbf{A}_k\|_F$, where $\mathbf{A}_k$ is the best rank-$k$ approximation to $\mathbf{A}$ and runs in $O(n\text{poly}(\frac{k}{\epsilon}))$.*

## 6 Experiments

In this section, we benchmark the performance of our sublinear time algorithm with the conventional SVD Algorithm (optimal error), iterative SVD methods and the input-sparsity time algorithm from [4]. We use the built-in svd function in numpy's linear algebra package to compute the truncated SVD. We also consider the iterative SVD algorithm implemented by the svds function in scipy's sparse linear algebra package, however, the error achieved is typically 3 orders of magnitude worse than computing the SVD and thus we defer the results to the Supplementary Material. We implement the input-sparsity time low-rank approximation algorithm from [4] using a count-sketch matrix [3].

**Synthetic Clustering Dataset**

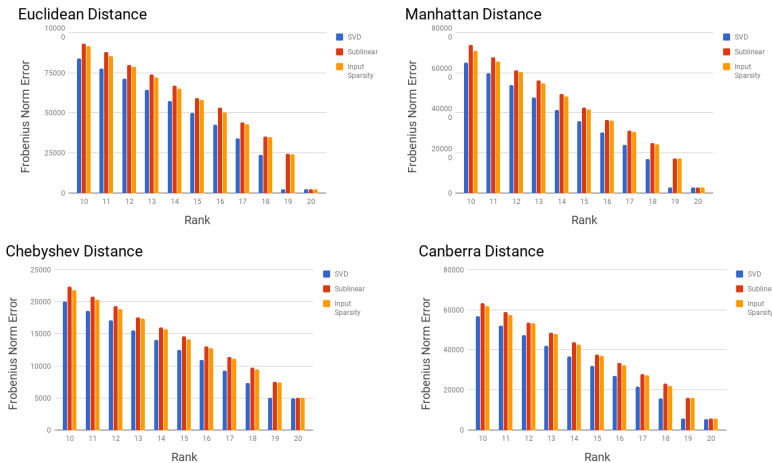

**MNIST Dataset**

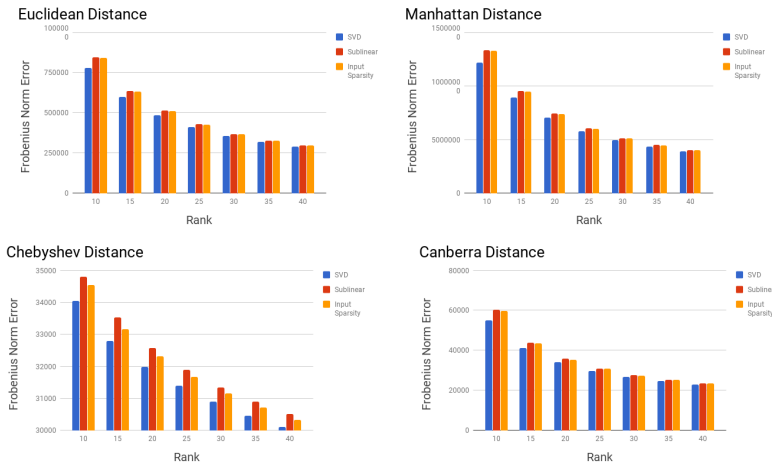

Figure 6.1: We plot error on a synthetic dataset with 20 clusters and the MNIST dataset. The distance matrix is created using $\ell_2$, $\ell_1$, $\ell_\infty$ and $\ell_c$ metrics. We compare the error achieved by SVD (optimal), our Sublinear Algorithm and the Input Sparsity Algorithm from [4].

Finally, we implement Algorithm 2, as this has small recursive overhead. The experiments are run on a Macbook Pro 2017 with 16GB RAM, 2.8GHz quad-core 7th-generation Intel Core i7 processor. The first dataset is a synthetic clustering dataset generated in scikit-learn using the *make blobs* function. We generate 10000 points with 200 features split up into 20 clusters. We note that given the clustered structure, this dataset is expected to have a good rank-20 approximation, as observed in our experiments. The second dataset we use is the popular MNIST dataset which is a collection of 70000 handwritten characters but sample 10000 points from it. In the Supplementary Material we also consider the Gisette and Poker datasets. Given $n$ points, $\{p_1, p_2, \ldots p_n\}$, in $\mathbf{R}^d$, we compute a $n \times n$ distance matrix $\mathbf{A}$, such that $\mathbf{A}_{i,j} = d(p_i, p_j)$, where $d$ is Manhattan ($\ell_1$), Euclidean ($\ell_2$), Chebyshev ($\ell_\infty$) or Canberra[3] ($\ell_c$) distance. We compare the Frobenius norm error of the algorithms in Figure 6.1 and their corresponding running time in Table 1 and 2. We note that our sublinear time algorithm is only marginally worse in terms of absolute error, but runs 100-250 times faster than SVD and 8-20 times faster than the input-sparsity time algorithm.

**Acknowledgments:** The authors thank partial support from the National Science Foundation under Grant No. CCF-1815840. Part of this work was done while the author was visiting the Simons Institute for the Theory of Computing.

## Footnotes

[1]See, e.g., `https://en.wikipedia.org/wiki/Distance_matrix`

[2]The full version of our paper is available at `https://arxiv.org/pdf/1809.06986.pdf`

[3]See `https://en.wikipedia.org/wiki/Canberra_distance`

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
