[Supplementary Material · nips_supplementary.pdf]

# A Appendix.

## A.1 Row and Column Norm Estimation

**Lemma 2.1.** *(Row Norm Estimation.) Let $\mathbf{A}$ be a $m \times n$ matrix such that $\mathbf{A}$ satisfies approximate triangle inequality. For $i \in [m]$ let $\mathbf{A}_{i,*}$ be the $i^{th}$ row of $\mathbf{A}$. Algorithm 1 uniformly samples $\Theta(b)$ elements from $\mathbf{A}_{i,*}$ and with probability at least $9/10$ outputs an estimator which obtains an $O\left(n/b\right)$-approximation to $\|\mathbf{A}_{i,*}\|_2^2$. Further, Algorithm 1 runs in $O(bm + n)$ time.*

*Proof.* It is easy to analyze the running time of Algorithm 1. Step 1 and 2 run in $O(n + m)$ as they correspond to reading a column and a row. Uniformly sampling $\Theta(b)$ indices for each row takes $O(bm)$ time. Overall, we get a running time of $O\left(bm + n\right)$.

By reading the first column of $\mathbf{A}$, we obtain a entry $\mathbf{A}_{x,1}$ such that it is the minimum entry of the first column, i.e. $x = \text{argmin}_{i \in [m]} \mathbf{A}_{i,1}$. Then, reading row $\mathbf{A}_{x,*}$, we obtain a entry $\mathbf{A}_{x,y}$ such that index $y = \text{argmax}_{j \in [n]} \mathbf{A}_{x,j}$ i.e., $\mathbf{A}_{x,y}$ is the largest entry in row $x$. Let $d$ denote the entry $\mathbf{A}_{x,y}$ of the input matrix. Further, let $d_{\max} = \max_{j,j' \in [n]} \max_{i \in [m]} |\mathbf{A}_{i,j} - \mathbf{A}_{i,j'}|$. Note, we cannot compute $d_{\max}$ without reading all the entries of the matrix and this is no longer sublinear time. We should think of $d_{\max}$ as representing the diameter of $\mathcal{Q}$ when $\mathbf{A}$ is a distance matrix. To see why this is true, observe, for $q_j, q_{j'} \in \mathcal{Q}$, $\max_{i \in [m]} |\mathbf{A}_{i,j} - \mathbf{A}_{i,j'}|$ represents the best lower bound on $d(q_j, q_{j'})$, via reverse triangle inequality. Finding the largest such lower bound over all pairs of points in $\mathcal{Q}$ is the best lower bound on the diameter of $\mathcal{Q}$.

Intuitively, we show that $d$ is a good proxy for $d_{\max}$. Recall, for row $\mathbf{A}_{i,*}$, Algorithm 1 outputs the estimator $\widetilde{X}_i = d^2 + \frac{n}{b} \sum_{\ell \in \mathcal{T}_i} \mathbf{A}_{i,\ell}^2$, where $\mathcal{T}_i$ is a uniform sample of indices in the range $[1, n]$, such that $|\mathcal{T}_i| = \Theta(b)$. Let us first consider the case of estimating the norm of $\mathbf{A}_{x,*}$. Note, Algorithm 1 reads the entire row and can compute the norm exactly. However, the analysis of our estimator is more intuitive in this case. Observe, by approximate triangle inequality and the definition of $d_{\max}$,

$$\mathbf{A}_{x,j} \leq (1 + \epsilon)\left(\mathbf{A}_{x,y} + \max_{i \in [m]} |\mathbf{A}_{i,y} - \mathbf{A}_{i,j}|\right) \leq (1 + \epsilon)(d + d_{\max}) \tag{A.1}$$

where first inequality follows from the upper bound in 2.1, the second follows from recalling that $\mathbf{A}_{x,y} = d$ and observing that $\max_{i \in [m]} |\mathbf{A}_{i,y} - \mathbf{A}_{i,j}|$ is upper bounded by $d_{\max}$. Similarly, using the lower bound in 2.1,

$$\mathbf{A}_{x,j} \geq \frac{|\mathbf{A}_{x,y} - \max_{i \in [m]} |\mathbf{A}_{i,y} - \mathbf{A}_{i,j}||}{1 + \epsilon} \geq \frac{|d - d_{\max}|}{(1 + \epsilon)} \tag{A.2}$$

Intuitively, if $d_{\max}$ is sufficiently larger than $d$, $\mathbf{A}_{x,j}$ is within a constant factor of $d_{\max}$. Formally, if $d_{\max} \geq 2d$, for all $j \in [n]$, $\frac{d_{\max}}{2(1+\epsilon)} \leq \mathbf{A}_{x,j} \leq \frac{3(1+\epsilon)d_{\max}}{2}$. Therefore, each entry in $\mathbf{A}_{x,*}$ is at least $d_{\max}/2(1 + \epsilon)$. Observe, by linearity of expectation, $\mathbb{E}[\widetilde{X}_x] = d^2 + \|\mathbf{A}_{x,*}\|_2^2$. Therefore, by Markov's bound we have that with probability at least $1 - 1/c$,

$$\widetilde{X}_x \leq c(d^2 + \|\mathbf{A}_{x,*}\|_2^2) \leq 5c \|\mathbf{A}_{x,*}\|_2^2 \tag{A.3}$$

where the last inequality follows from $\|\mathbf{A}_{x,*}\|_2^2 \geq \frac{nd^2}{4(1+\epsilon)^2}$ (since each entry of $\mathbf{A}_{x,*}$ is at least $d/2(1 + \epsilon)$). Further,

$$\widetilde{X}_x \geq d^2 + \left(\frac{n}{b}\right)\left(\frac{bd^2}{4(1+\epsilon)^2}\right) \geq \frac{\|\mathbf{A}_{x,*}\|_2^2}{160} \tag{A.4}$$

where the first inequality follows from each sampled entry being at least $\frac{d}{2(1+\epsilon)}$ and the second inequality follows from $\|\mathbf{A}_{x,*}\|_2^2 \leq \left(\frac{3(1+\epsilon)d}{2}\right)^2 n \leq 10nd^2$. For an appropriate value of $c$, $\widetilde{X}_x = \Theta\left(\|\mathbf{A}_{x,*}\|_2^2\right)$ with probability at least $99/100$.

Note, $d$ could be really small compared to $d_{\max}$ and we cannot hope for a constant approximation to the row norm. Formally, $d \leq d_{\max}$ and we consider two cases, one where $\|\mathbf{A}_{x,*}\|_2^2 \leq \frac{nd^2}{b}$ and the

second being its complement. In the first case, we observe,

$$\widetilde{X}_x \geq d^2 \geq \frac{b}{n} \|\mathbf{A}_{x,*}\|_2^2 \tag{A.5}$$

where the first inequality follows from $\|\mathbf{A}_{x,*}\|_2^2 \geq 0$ and the second follows from $\|\mathbf{A}_{x,*}\|_2^2 \leq \frac{nd^2}{b}$. On the other hand, observe, by linearity of expectation, $\mathbb{E}[\widetilde{X}_x] = d^2 + \|\mathbf{A}_{x,*}\|_2^2$. By Markov's bound, with probability at least $1 - \frac{1}{c}$,

$$\widetilde{X}_x \leq c(d^2 + \|\mathbf{A}_{x,*}\|_2^2) \leq 2c \|\mathbf{A}_{x,*}\|_2^2 \tag{A.6}$$

where the last inequality follows from $d$ being an element of $\mathbf{A}_{x,*}$. Therefore, combining the upper and lower bound, with probability at least $99/100$, $\widetilde{X}_x$ achieves a $\Theta\left(\frac{n}{b}\right)$-approximation to $\|\mathbf{A}_{x,*}\|_2^2$.

Next, consider the second case where $\|\mathbf{A}_{x,*}\|_2^2 \geq \frac{nd^2}{b}$. We begin by bounding $\mathbf{Var}[\widetilde{X}_x]$. Let $Y_{x,j} = \mathbf{A}_{x,j}^2$ with probability $b/n$, and $Y_{x,j} = 0$ otherwise. Then,

$$
\begin{aligned}
\mathbf{Var}[\widetilde{X}_x] = \mathbf{Var}\left[d^2 + \frac{n}{b}\sum_{j \in \mathcal{T}_x} Y_{x,j}\right] &= \left(\frac{n}{b}\right)^2 \mathbf{Var}\left[\sum_{j \in \mathcal{T}_x} Y_{x,j}\right] \\
&\leq \left(\frac{n}{b}\right)^2 \mathbb{E}\left[(\sum_{j \in \mathcal{T}_x} Y_{x,j})^2\right] \\
&\leq \left(\frac{n}{b}\right)^2 \frac{b}{n}\left(\frac{\|\mathbf{A}_{x,*}\|_2^2}{d^2}\right) d^4 \\
&\leq \frac{\|\mathbf{A}_{x,*}\|_2^4}{100c}
\end{aligned}
\tag{A.7}
$$

where $c$ is another fixed constant, the first inequality follows from the definition of variance, the second follows from $Y_{x,j}$ being upper bounded by $d^2$ and an averaging argument, and the last follows from the assumption that $\frac{nd^2}{b} \leq \|\mathbf{A}_{x,*}\|_2^2$. Observe that the variance is maximized when there are $\frac{\|\mathbf{A}_{x,*}\|_2^2}{d^2}$ entries with value $d$ and the last inequality follows from our assumption in this case. By Chebyshev's inequality,

$$\mathbf{Pr}\left[\left|\widetilde{X}_x - \mathbb{E}\left[\widetilde{X}_x\right]\right| > \frac{\|\mathbf{A}_{x,*}\|_2^2}{100}\right] \leq \frac{\mathbf{Var}\left[\widetilde{X}_x\right]}{c^2 \|\mathbf{A}_{x,*}\|_2^4} \leq \frac{1}{c^2} \tag{A.8}$$

Therefore, with probability at least $1 - 1/c^2$,

$$\widetilde{X}_x = \mathbb{E}[\widetilde{X}_x] \pm \frac{\|\mathbf{A}_{x,*}\|_2^2}{100} = d^2 + \left(1 \pm \frac{1}{100}\right) \|\mathbf{A}_{x,*}\|_2^2 \tag{A.9}$$

Therefore, $\widetilde{X}_x$ achieves a $\Theta(1)$-approximation to $\|\mathbf{A}_{x,*}\|_2^2$. It follows that in every case, $\widetilde{X}_x$ achieves an $O\left(\frac{n}{b}\right)$-approximation to $\|\mathbf{A}_{x,*}\|_2^2$ with probability at least $9/10$.

Now we analyze our estimator in its full generality by considering row $\|\mathbf{A}_{i,*}\|_2^2$ for any $i \neq x$. We define $d' = \max_{j \in [n]} \mathbf{A}_{i,j}$ to be the largest entry in row $i$. Note, we do not explicitly know $d'$ as this would require reading the entire row. Instead, we show that biasing our estimator with $d^2$ suffices for all rows. Recall, $\mathbf{A}_{x,y} = \max_{j \in [n]} \mathbf{A}_{x,j}$. We follow an analysis similar to the simplified one above. Since our estimator is still biased by $d^2$, we analyze the cases where $d$ is small or large compared to $d_{\max}$ separately. Consider the first case, where $d \geq 8d_{\max}$. We begin by bounding $d'$ in terms of $d$ and $d_{\max}$. By the approximate triangle inequality,

$$\mathbf{A}_{x,1} \geq \frac{|\mathbf{A}_{x,y} - \max_{i \in [m]} |\mathbf{A}_{i,1} - \mathbf{A}_{i,y}||}{1+\epsilon} \geq \frac{|d - d_{\max}|}{1+\epsilon} \geq \frac{7}{1+\epsilon} d_{\max} \geq 3d_{\max} \tag{A.10}$$

where the second inequality follows from recalling the definition of $d$ and observing that $d_{\max} \geq \max_{i \in [m]} |\mathbf{A}_{i,1} - \mathbf{A}_{i,y}|$, and the third inequality follows from the assumption in this case. Alternatively, we can repeat the above bound to get

$$\mathbf{A}_{x,1} \geq \frac{|\mathbf{A}_{x,y} - \max_{i \in [m]} |\mathbf{A}_{i,1} - \mathbf{A}_{i,y}||}{1+\epsilon} \geq \frac{|d - d_{\max}|}{1+\epsilon} \geq \frac{7d}{(1+\epsilon)8} \geq \frac{7d}{16} \tag{A.11}$$

Further,

$$\mathbf{A}_{x,1} \leq \mathbf{A}_{i,1} \leq d' \tag{A.12}$$

where the first inequality follows from the definition of $\mathbf{A}_{x,1}$ and the second inequality follows from the definition of $d'$. Therefore, combining the two equations above, we get that $d' \geq 3d_{\max}$ and $d' \geq \frac{7d}{16}$. Next, we show a bound of any entry of the $i$-th row. By the upper bound in approximate triangle inequality,

$$\mathbf{A}_{i,j} \leq (1+\epsilon)\left(\mathbf{A}_{i,j'} + \max_{i'} |\mathbf{A}_{i',j} - \mathbf{A}_{i',j'}|\right) \leq (1+\epsilon)\left(d' + d_{\max}\right) \leq \frac{8d'}{3} \tag{A.13}$$

where the second inequality follows from $d'$ being the largest element in row $i$, and the definition of $d_{\max}$, and the last follows from $d' \geq 3d_{\max}$. Similarly, by the lower bound in approximate triangle inequality,

$$\mathbf{A}_{i,j} \geq \frac{|\mathbf{A}_{i,j'} - \max_{i'} |\mathbf{A}_{i',j} - \mathbf{A}_{i',j'}||}{1+\epsilon} \geq \frac{|d' - d_{\max}|}{1+\epsilon} \geq \frac{d'}{3} \tag{A.14}$$

Combining the two equations above, for all $j \in [n]$, $\mathbf{A}_{i,j} = \Theta(d')$, all entries of $\mathbf{A}_{i,*}$ are within a constant factor of each other. Therefore, $\|\mathbf{A}_{i,*}\|_2^2 \leq \frac{64nd'^2}{9}$. Recall, our estimator is

$$\widetilde{X}_i = d^2 + \sum_{j \in \mathcal{T}_i} \frac{n}{b} \mathbf{A}_{i,j}^2 \geq \frac{n}{b} \frac{bd'^2}{9} \geq \frac{\|\mathbf{A}_{i,*}\|_2^2}{64} \tag{A.15}$$

We observe that by linearity of expectation, $\mathbb{E}[\widetilde{X}_i] = d^2 + \|\mathbf{A}_{i,*}\|_2^2$. By Markov's we know that with probability at least $1 - 1/c$,

$$\widetilde{X}_i \leq c\left(d^2 + \|\mathbf{A}_{i,*}\|_2^2\right) \leq c\left(\frac{216}{49}d'^2 + \|\mathbf{A}_{i,*}\|_2^2\right) \leq c'\|\mathbf{A}_{i,*}\|_2^2 \tag{A.16}$$

where the second inequality follows from recalling that $d' \geq \frac{7d}{16}$ and the last inequality follows from $d'$ being an entry in $\mathbf{A}_{i,*}$. Therefore, $\widetilde{X}_i = \Theta\left(\|\mathbf{A}_{i,*}\|_2^2\right)$, with probability at least $99/100$.

Now we analyze the case where $d \leq 8d_{\max}$. We then consider two cases: $\|\mathbf{A}_{i,*}\|_2^2 \geq \frac{nd'^2}{b}$ or $\|\mathbf{A}_{i,*}\|_2^2 \leq \frac{nd'^2}{b}$. In the first case, computing the variance exactly as in equation A.7 and applying Chebyshev's inequality, we get that with probability at least $1 - 1/c^2$,

$$\widetilde{X}_i = d^2 + \left(1 \pm \frac{1}{100}\right) \|\mathbf{A}_{i,*}\|_2^2 \tag{A.17}$$

Since $d \leq 8d_{\max}$ and $\frac{1}{16}d_{\max}^2 \leq \|\mathbf{A}_{i,*}\|_2^2$, $\widetilde{X}_i = \Theta\left(\|\mathbf{A}_{i,*}\|_2^2\right)$ with probability $99/100$.

In the second case, we show that all entries of $\mathbf{A}_{i,*}$ are within a constant factor of each other. Recall, $\widetilde{X}_i \geq d^2$ since $\|\mathbf{A}_{i,*}\|_2^2 \geq 0$. If $8d \geq d'$,

$$\widetilde{X}_i \geq \frac{d'^2}{64} \geq \frac{b\|\mathbf{A}_{i,*}\|_2^2}{64n} \tag{A.18}$$

Recall, $\mathbb{E}[\widetilde{X}_i] = d^2 + \|\mathbf{A}_{i,*}\|_2^2$, $d \leq \frac{16}{7}d'$ and $d'^2 \leq \|\mathbf{A}_{i,*}\|_2^2$. Therefore, the upper bound on $\widetilde{X}_i$ follows from Markov's bound and holds with probability at least $99/100$. If instead $8d \leq d'$, we show all entries of $\mathbf{A}_{i,*}$ are within a constant factor of each other. To see this, let $d' = \mathbf{A}_{i,j^*}$ and observe by approximate triangle inequality,

$$\mathbf{A}_{i,j} \leq (1+\epsilon)\left(\mathbf{A}_{i,j^*} + \max_{i'} |\mathbf{A}_{i',j} - \mathbf{A}_{i',j^*}|\right) \leq 2d' + 2(1+\epsilon)(\mathbf{A}_{x,j^*} + \mathbf{A}_{x,j})$$
$$\leq 2d' + 8d \leq 3d' \tag{A.19}$$

where the first and second inequalities follow from initially applying the upper bound in equation 2.1 to $\mathbf{A}_{i,j}$, and then applying the equation 2.2 to $\max_{i'} |\mathbf{A}_{i',j} - \mathbf{A}_{i',j^*}|$, the third inequality follows from upper bounding both $\mathbf{A}_{x,i^*}$ and $\mathbf{A}_{x,j}$ by $d$, and the last inequality follows from the assumption $8d \leq d'$ combined with the definition of $d'$. Further,

$$\begin{aligned}
\mathbf{A}_{i,j} \geq \frac{\mathbf{A}_{i,i^*} - \mathbf{A}_{i^*j}}{1+\epsilon} &\geq \frac{1}{2}(d' - 2(\mathbf{A}_{x,j^*} + \mathbf{A}_{x,j})) \\
&\geq \frac{1}{2}d' - 2d \geq \frac{d'}{4}
\end{aligned} \tag{A.20}$$

where the third inequality follows from upper bounding both $\mathbf{A}_{x,i^*}$ and $\mathbf{A}_{x,j}$ by $d$, and the last inequality follows from the assumption $8d \leq d'$ combined with the definition of $d'$. Therefore, $d(p_i, q_j) = \Theta(d')$. Finally, $\mathbb{E}[\widetilde{X}_i] = d^2 + \|\mathbf{A}_{i,*}\|_2^2$ and by Markov's bound, our estimator $\widetilde{X}_i = \Theta\left(d^2 + \|\mathbf{A}_{i,*}\|_2^2\right)$ with probability $99/100$. Since $d^2 \leq \|\mathbf{A}_{i,*}\|_2^2$, we obtain a constant factor approximation with probability at least $99/100$. By a union bound over all the probabilistic events, all of them simultaneously hold with probability at least $9/10$, which finishes the proof. $\square$

To obtain an $O\left(\frac{n}{b}\right)$ approximation for all the $m$ rows simultaneously with constant probability, we can compute $O\left(\log(m)\right)$ estimators for each row and take their median. We also observe that Column and Row Norm Estimation are symmetric operations and a slight modification to Algorithm 1 yields a Column Norm Estimation algorithm with the following guarantee:

**Corollary A.1.** *(Column Norm Estimation.) Let $\mathbf{A}$ be a $m \times n$ matrix such that $\mathbf{A}$ satisfies approximate triangle inequality. For $j \in [n]$ let $\mathbf{A}_{*,j}$ be the $j^{th}$ column of $\mathbf{A}$. There exists an algorithm that uniformly samples $\Theta(b)$ elements from $\mathbf{A}_{*,j}$ and with probability $9/10$ outputs an estimator which is an $O\left(\frac{m}{b}\right)$-approximation to $\|\mathbf{A}_{*,j}\|_2^2$. Further, this algorithm runs in $O(bn + m)$ time.*

## A.2 Projection-Cost Preserving Sketches

Next, we describe how to use the above estimators to sample rows and columns. We would like to reduce the dimensionality of the matrix in a way that approximately preserves low-rank structure. At a high level, we sketch the input matrix on the left and the right and use an input-sparsity time algorithm on the resulting smaller matrix. The main insight to show such a result is that if we can approximately preserve all rank-$k$ subspaces in the column and row space of the matrix, then we can recursively sample rows and columns to obtain a much smaller matrix. To this end, we introduce a relaxation of projection-cost preserving sketches [6] that satisfy an *additive error guarantee*.

We prove that projection-cost preserving sketches can be computed using our coarse estimates to the column and row norms. We begin by showing an intermediate result bounding the spectrum of the sample in terms of the original matrix.

**Theorem A.1.** *(Spectral Bounds.) Let $\mathbf{A}$ be a $m \times n$ matrix such that $\mathbf{A}$ satisfies approximate triangle inequality. For $j \in [n]$, let $\widetilde{X}_j$ be an $O\left(\frac{m}{b}\right)$-approximate estimate for the $j^{th}$ column of $\mathbf{A}$ such that it satisfies the guarantee of Corollary A.1. Then, let $q = \{q_1, q_2 \ldots q_n\}$ be a probability distribution over the columns of $\mathbf{A}$ such that $q_j = \frac{\widetilde{X}_j}{\sum_{j'} \widetilde{X}_{j'}}$. Let $t = O\left(\frac{m}{b\epsilon^2} \log(\frac{m}{\delta})\right)$ for some constant $c$. Then, construct $\mathbf{C}$ using $t$ columns of $\mathbf{A}$ and set each one to $\frac{\mathbf{A}_{*,j}}{\sqrt{tq_j}}$ with probability $q_j$. With probability at least $1 - \delta$,*

$$\mathbf{C}\mathbf{C}^T - \epsilon \|\mathbf{A}\|_F^2 \mathbb{I} \preceq \mathbf{A}\mathbf{A}^T \preceq \mathbf{C}\mathbf{C}^T + \epsilon \|\mathbf{A}\|_F^2 \mathbb{I}$$

*Proof.* Let $\mathbf{Y} = \mathbf{C}\mathbf{C}^T - \mathbf{A}\mathbf{A}^T$. For notational convenience let $\mathbf{A}_j = \mathbf{A}_{*,j}$. We can then write $\mathbf{Y} = \sum_{i \in [t]} \mathbf{X}_i$, where $\mathbf{X}_i = \frac{1}{t}(\frac{1}{q_j}\mathbf{A}_j\mathbf{A}_j^T - \mathbf{A}\mathbf{A}^T)$ with probability $q_j$. We observe that $\mathbb{E}[\frac{1}{q_j}\mathbf{A}_j\mathbf{A}_j^T - \mathbf{A}\mathbf{A}^T] = 0$, and therefore, $\mathbb{E}[\mathbf{Y}] = 0$. Next, we bound the operator norm of $\mathbf{Y}$. To this end, we use the Matrix Bernstein inequality, which in turn requires a bound on the operator norm of $\mathbf{X}_i$ and

variance of $\mathbf{Y}$. Recall,

$$
\begin{aligned}
\|\mathbf{X}_i\|_2 &= \left\|\frac{1}{tq_j}\mathbf{A}_j\mathbf{A}_j^T - \frac{1}{t}\mathbf{A}\mathbf{A}^T\right\|_2 \\
&\leq \frac{m}{tb}\frac{\|\mathbf{A}\|_F^2}{\|\mathbf{A}_j\|_2^2}\|\mathbf{A}_j\mathbf{A}_j^T\|_2 + \frac{1}{t}\|\mathbf{A}\mathbf{A}^T\|_2 \qquad\text{(A.21)} \\
&\leq \frac{2m}{tb}\|\mathbf{A}\|_F^2
\end{aligned}
$$

Next, we bound $\mathbf{Var}[\mathbf{Y}] \leq \mathbb{E}\left[\mathbf{Y}^2\right]$.

$$
\begin{aligned}
\mathbb{E}\left[\mathbf{Y}^2\right] = t\mathbb{E}\left[\mathbf{X}_i^2\right] &= \frac{1}{t}\mathbb{E}\left[\left(\frac{1}{q_j}\mathbf{A}_j\mathbf{A}_j^T - \mathbf{A}\mathbf{A}^T\right)^2\right] \\
&= \frac{1}{t}\left(\frac{(\mathbf{A}_j\mathbf{A}_j^T)^2}{q_j} + q_j(\mathbf{A}\mathbf{A}^T)^2 - 2\mathbf{A}_j\mathbf{A}_j^T\mathbf{A}\mathbf{A}^T\right) \\
&\preceq \frac{1}{t}\left(\frac{m}{b}\frac{\|\mathbf{A}\|_F^2}{\|\mathbf{A}_j\|_2^2}(\mathbf{A}_j\mathbf{A}_j^T)^2 + \frac{b}{m}\frac{\|\mathbf{A}_j\|_2^2}{\|\mathbf{A}\|_F^2}(\mathbf{A}\mathbf{A}^T)^2\right) \qquad\text{(A.22)} \\
&\preceq \frac{cm\|\mathbf{A}\|_F^4}{tb}\mathbb{I}_{m\times m}
\end{aligned}
$$

Therefore, $\sigma^2 = \|\mathbb{E}\left[\mathbf{Y}^2\right]\|_2 \leq \frac{cm\|\mathbf{A}\|_F^4}{tb}$ Applying the Matrix Bernstein inequality (see Lemma **??** in the Appendix),

$$
\mathbf{Pr}\left[\|\mathbf{Y}\|_2 \geq \epsilon\|\mathbf{A}\|_F^2\right] \leq me^{\left(-\frac{\epsilon^2\|\mathbf{A}\|_F^4}{\sigma^2 + \frac{\epsilon m}{3tb}\|\mathbf{A}\|_F^4}\right)} \leq \delta
$$

by substituting the value of $\sigma^2$ and setting $t = \frac{c'm}{b\epsilon^2}\log(\frac{m}{\delta})$. The bound follows. $\qquad\square$

Using the above theorem, we show that sampling columns of $\mathbf{A}$ according to approximate column norms yields a matrix that preserves projection cost for all rank-$k$ projections up to additive error.

**Theorem A.2.** *(Column Projection-Cost Preservation.) Let $\mathbf{A}$ be a $m\times n$ matrix such that $\mathbf{A}$ satisfies approximate triangle inequality. For $j \in [n]$, let $\widetilde{X}_j$ be an $O\left(\frac{m}{b}\right)$-approximate estimate for the $j^{th}$ column of $\mathbf{A}$ such that it satisfies the guarantee of Corollary A.1. Then, let $q = \{q_1, q_2 \ldots q_n\}$ be a probability distribution over the columns of $\mathbf{A}$ such that $q_j = \frac{\widetilde{X}_j}{\sum_{j'}\widetilde{X}_{j'}}$. Let $t = O\left(\frac{mk^2}{b\epsilon^2}\log(\frac{m}{\delta})\right)$ for some constant c. Then, construct $\mathbf{C}$ using $t$ columns of $\mathbf{A}$ and set each one to $\frac{A_{*,j}}{\sqrt{tq_j}}$ with probability $q_j$. With probability at least $1 - \delta$, for any rank-$k$ orthogonal projection $\mathbf{X}$*

$$
\|\mathbf{C} - \mathbf{X}\mathbf{C}\|_F^2 = \|\mathbf{A} - \mathbf{X}\mathbf{A}\|_F^2 \pm \epsilon\|\mathbf{A}\|_F^2
$$

*Proof.* We give a similar proof to the relative error guarantees in [6], but need to replace certain parts with our different distribution which is only based on row and column norms rather than leverage scores, and consequently we obtain additive error in places instead. As our lower bound shows, this is necessary in our setting.

Let $\mathbf{Y} = \mathbb{I} - \mathbf{X}$, so that $\|\mathbf{A} - \mathbf{X}\mathbf{A}\|_F^2 = \text{Tr}\left(\mathbf{Y}\mathbf{A}\mathbf{A}^T\mathbf{Y}\right)$ and $\|\mathbf{C} - \mathbf{X}\mathbf{C}\|_F^2 = \text{Tr}\left(\mathbf{Y}\mathbf{C}\mathbf{C}^T\mathbf{Y}\right)$. We split the singular values of $\mathbf{A}$ into a head and a tail as follows. Let $\sigma_\ell^2$ be the smallest singular value of $\mathbf{A}$ for which $\sigma_\ell^2 \geq \frac{\|\mathbf{A}\|_F^2}{k}$. Let $\mathbf{U}_\ell\mathbf{U}_\ell^T$ be the projection onto the top $\ell$ singular vectors of $\mathbf{A}$ and $\mathbf{U}_{\backslash\ell}\mathbf{U}_{\backslash\ell}^T$ be the projection on the bottom singular vectors. Let $\mathbf{P}_\ell = \mathbf{U}_\ell\mathbf{U}_\ell^T$ and $\mathbf{P}_{\backslash\ell} = \mathbf{U}_{\backslash\ell}\mathbf{U}_{\backslash\ell}^T$. Then,

$$
\begin{aligned}
\text{Tr}\left(\mathbf{Y}\mathbf{A}\mathbf{A}^T\mathbf{Y}\right) &= \text{Tr}\left(\mathbf{Y}\mathbf{P}_\ell\mathbf{A}\mathbf{A}^T\mathbf{P}_\ell\mathbf{Y}\right) + \text{Tr}\left(\mathbf{Y}\mathbf{P}_{\backslash\ell}\mathbf{A}\mathbf{A}^T\mathbf{P}_{\backslash\ell}\mathbf{Y}\right) + 2\text{Tr}\left(\mathbf{Y}\mathbf{P}_\ell\mathbf{A}\mathbf{A}^T\mathbf{P}_{\backslash\ell}\mathbf{Y}\right) \\
&= \text{Tr}\left(\mathbf{Y}\mathbf{P}_\ell\mathbf{A}\mathbf{A}^T\mathbf{P}_\ell\mathbf{Y}\right) + \text{Tr}\left(\mathbf{Y}\mathbf{P}_{\backslash\ell}\mathbf{A}\mathbf{A}^T\mathbf{P}_{\backslash\ell}\mathbf{Y}\right)
\end{aligned}
$$
$$\text{(A.23)}$$

The cross terms vanish since $\mathbf{P}_\ell\mathbf{A}$ and $\mathbf{P}_{\backslash\ell}\mathbf{A}$ are orthogonal. Similarly, we split the $\mathbf{C}\mathbf{C}^T$ terms.

$$
\text{Tr}\left(\mathbf{Y}\mathbf{C}\mathbf{C}^T\mathbf{Y}\right) = \text{Tr}\left(\mathbf{Y}\mathbf{P}_\ell\mathbf{C}\mathbf{C}^T\mathbf{P}_\ell\mathbf{Y}\right) + \text{Tr}\left(\mathbf{Y}\mathbf{P}_{\backslash\ell}\mathbf{C}\mathbf{C}^T\mathbf{P}_{\backslash\ell}\mathbf{Y}\right) + 2\text{Tr}\left(\mathbf{Y}\mathbf{P}_\ell\mathbf{C}\mathbf{C}^T\mathbf{P}_{\backslash\ell}\mathbf{Y}\right)
$$
$$\text{(A.24)}$$

We note that the cross terms here do not vanish since $\mathbf{P}_\ell \mathbf{C}$ and $\mathbf{P}_{\backslash\ell}\mathbf{C}$ might not be orthogonal. We now show how to handle each of these terms separately.

## A.3 Head Terms.

Our analysis for the Head Terms closely follows that of [6], where they strive for relative error using leverage scores instead. For any vector $x$, let $y = \mathbf{P}_\ell x$. Then, $y^T \mathbf{A}\mathbf{A}^T y = x^T \mathbf{P}_\ell^T \mathbf{A}\mathbf{A}^T \mathbf{P}_\ell x = x^T \mathbf{A}_\ell \mathbf{A}_\ell^T x$. Then, setting $\epsilon = \frac{\epsilon}{k}$ in Theorem $A.1$, we obtain

$$y^T \mathbf{C}\mathbf{C}^T y - \frac{\epsilon\|\mathbf{A}\|_F^2}{k} y^T y \le x^T \mathbf{A}_\ell \mathbf{A}_\ell^T x \le y^T \mathbf{C}\mathbf{C}^T y + \frac{\epsilon\|\mathbf{A}\|_F^2}{k} y^T y \qquad (A.25)$$

Note, this only increases the number of columns we sample by a poly($k$) factor. Recall, by definition, $y$ is orthogonal to all but the top $\ell$ singular vectors of $\mathbf{A}$. Therefore, $x^T \mathbf{A}_\ell \mathbf{A}_\ell^T x = y^T \mathbf{A}\mathbf{A}^T y \ge \frac{\|\mathbf{A}\|_F^2}{k} y^T y$. Combined with $(A.25)$, $y^T \mathbf{C}\mathbf{C}^T y = (1 \pm \epsilon)x^T \mathbf{A}_\ell \mathbf{A}_\ell^T x$. Since $y^T \mathbf{C}\mathbf{C}^T y = x^T \mathbf{P}_\ell \mathbf{C}\mathbf{C}^T \mathbf{P}_\ell x$ and the above is true for any $x$, we get

$$(1 - \epsilon)\mathbf{P}_\ell \mathbf{C}\mathbf{C}^T \mathbf{P}_\ell \preceq \mathbf{A}_\ell \mathbf{A}_\ell^T \preceq (1 + \epsilon)\mathbf{P}_\ell \mathbf{C}\mathbf{C}^T \mathbf{P}_\ell \qquad (A.26)$$

We observe that $(A.26)$ bounds the diagonal entries of $\mathbf{Y}\mathbf{A}_\ell \mathbf{A}_\ell^T \mathbf{Y}$ in terms of the diagonal entries of $\mathbf{Y}\mathbf{P}_\ell \mathbf{C}\mathbf{C}^T \mathbf{P}_\ell \mathbf{Y}$, we get that,

$$(1 - \epsilon)\mathrm{Tr}\left(\mathbf{Y}\mathbf{P}_\ell \mathbf{C}\mathbf{C}^T \mathbf{P}_\ell \mathbf{Y}\right) \le \mathrm{Tr}\left(\mathbf{Y}\mathbf{P}_\ell \mathbf{A}\mathbf{A}^T \mathbf{P}_\ell \mathbf{Y}\right) \le (1 + \epsilon)\mathrm{Tr}\left(\mathbf{Y}\mathbf{P}_\ell \mathbf{C}\mathbf{C}^T \mathbf{P}_\ell \mathbf{Y}\right)$$

Rearranging the terms and assuming $\epsilon < 1/2$,

$$(1 - 4\epsilon)\mathrm{Tr}\left(\mathbf{Y}\mathbf{P}_\ell \mathbf{A}\mathbf{A}^T \mathbf{P}_\ell \mathbf{Y}\right) \le \mathrm{Tr}\left(\mathbf{Y}\mathbf{P}_\ell \mathbf{C}\mathbf{C}^T \mathbf{P}_\ell \mathbf{Y}\right) \le (1 + 4\epsilon)\mathrm{Tr}\left(\mathbf{Y}\mathbf{P}_\ell \mathbf{A}\mathbf{A}^T \mathbf{P}_\ell \mathbf{Y}\right) \quad (A.27)$$

## A.4 Tail Terms.

Recall from the definition of $\mathbf{Y}$,

$$\mathrm{Tr}\left(\mathbf{Y}\mathbf{A}_{\backslash\ell}\mathbf{A}_{\backslash\ell}^T \mathbf{Y}\right) = \mathrm{Tr}\left(\mathbf{A}_{\backslash\ell}\mathbf{A}_{\backslash\ell}^T\right) - \mathrm{Tr}\left(\mathbf{X}\mathbf{A}_{\backslash\ell}\mathbf{A}_{\backslash\ell}^T \mathbf{X}\right) \qquad (A.28)$$

Similarly,

$$\mathrm{Tr}\left(\mathbf{Y}\mathbf{P}_{\backslash\ell}\mathbf{C}\mathbf{C}^T \mathbf{P}_{\backslash\ell}\mathbf{Y}\right) = \mathrm{Tr}\left(\mathbf{P}_{\backslash\ell}\mathbf{C}\mathbf{C}^T \mathbf{P}_{\backslash\ell}\right) - \mathrm{Tr}\left(\mathbf{X}\mathbf{P}_{\backslash\ell}\mathbf{C}\mathbf{C}^T \mathbf{P}_{\backslash\ell}\mathbf{X}\right) \qquad (A.29)$$

In order to relate $\mathrm{Tr}\left(\mathbf{P}_{\backslash\ell}\mathbf{C}\mathbf{C}^T \mathbf{P}_{\backslash\ell}\right)$ and $\mathrm{Tr}\left(\mathbf{A}_{\backslash\ell}\mathbf{A}_{\backslash\ell}^T\right)$, we observe that the first term is equal to the second in expectation. Therefore, using a scalar Chernoff bound, we show that $\mathrm{Tr}\left(\mathbf{P}_{\backslash\ell}\mathbf{C}\mathbf{C}^T \mathbf{P}_{\backslash\ell}\right)$ concentrates around its expectation. We defer this proof to the Supplementary Material and obtain the following bound:

$$\mathrm{Tr}\left(\mathbf{A}_{\backslash\ell}\mathbf{A}_{\backslash\ell}^T\right) - \mathrm{Tr}\left(\mathbf{P}_{\backslash\ell}\mathbf{C}\mathbf{C}^T \mathbf{P}_{\backslash\ell}\right) = \pm\epsilon\|\mathbf{A}\|_F^2 \qquad (A.30)$$

Next, we relate the remaining two terms following a strategy similar to the one used for the head terms. Let the vectors $x, y$ be defined as above. Then, $x^T \mathbf{A}_{\backslash\ell}\mathbf{A}_{\backslash\ell}^T x = y^T \mathbf{A}\mathbf{A}^T y$. Using Theorem A.1 with $\epsilon = \frac{\epsilon}{k}$, we get

$$x^T \mathbf{A}_{\backslash\ell}\mathbf{A}_{\backslash\ell}^T x = y^T \mathbf{C}\mathbf{C}^T y \pm \frac{\epsilon\|\mathbf{A}\|_F^2}{k} y^T y \qquad (A.31)$$

Since $\mathbf{P}_{\backslash\ell}$ is a projection matrix, $y^T y \le x^T x$ and assuming $\epsilon < 1/2$,

$$y^T \mathbf{C}\mathbf{C}^T y - \epsilon\frac{\|A\|_F^2}{k} x^T x \le x^T \mathbf{A}_{\backslash\ell}\mathbf{A}_{\backslash\ell}^T x$$

$$x^T \mathbf{A}_{\backslash\ell}\mathbf{A}_{\backslash\ell}^T x \le y^T \mathbf{C}\mathbf{C}^T y + \epsilon\frac{\|\mathbf{A}\|_F^2}{k} x^T x \qquad (A.32)$$

Recall, by the definition of $m$, $x^T \mathbf{A}_{\backslash\ell}\mathbf{A}_{\backslash\ell}^T x \le \frac{\|A\|_F^2}{k}$. Substituting this back into $(A.32)$ and $(A.33)$, we get

$$P_{\backslash\ell}\mathbf{C}\mathbf{C}^T \mathbf{P}_{\backslash\ell} - \epsilon\frac{\|\mathbf{A}\|_F^2}{k}\mathbb{I} \preceq \mathbf{A}_{\backslash\ell}\mathbf{A}_{\backslash\ell}^T \preceq \mathbf{P}_{\backslash\ell}\mathbf{C}\mathbf{C}^T \mathbf{P}_{\backslash\ell} + \epsilon\frac{\|\mathbf{A}\|_F^2}{k}\mathbb{I} \qquad (A.33)$$

Let $\mathbf{X} = \mathbf{Z}\mathbf{Z}^T$ such that $\mathbf{Z} \in \mathbb{R}^{m \times k}$ is an orthonomal matrix. By the cyclic property of the trace,

$$\mathrm{Tr}\left(\mathbf{X}\mathbf{A}_{\backslash \ell}\mathbf{A}_{\backslash \ell}^T\mathbf{X}\right) = \mathrm{Tr}\left(\mathbf{Z}^T\mathbf{A}_{\backslash \ell}\mathbf{A}_{\backslash \ell}^T\mathbf{Z}\right) = \sum_{j \in [k]} \mathbf{Z}_{*,j}^T\mathbf{A}_{\backslash \ell}\mathbf{A}_{\backslash \ell}^T\mathbf{Z}_{*,j}$$

Similarly,

$$\mathrm{Tr}\left(\mathbf{X}\mathbf{C}_{\backslash \ell}\mathbf{C}_{\backslash \ell}^T\mathbf{X}\right) \leq \sum_{j \in [k]} \mathbf{Z}_{*,j}^T\mathbf{C}_{\backslash \ell}\mathbf{C}_{\backslash \ell}^T\mathbf{Z}_{*,j}$$

Combining this with $(A.33)$ and $(A.31)$, and assuming $\epsilon < 1/2$ we get,

$$\mathrm{Tr}\left(\mathbf{Y}\mathbf{P}_{\backslash \ell}\mathbf{C}\mathbf{C}^T\mathbf{P}_{\backslash \ell}\mathbf{Y}\right) = \mathrm{Tr}\left(\mathbf{Y}\mathbf{A}_{\backslash \ell}\mathbf{A}_{\backslash \ell}^T\mathbf{Y}\right) \pm 4\epsilon\|A\|_F^2 \tag{A.34}$$

## A.5 Cross Terms.

Finally, we consider the cross term $2\mathrm{Tr}\left(\mathbf{Y}\mathbf{P}_\ell\mathbf{C}\mathbf{C}^T\mathbf{P}_{\backslash \ell}\mathbf{Y}\right)$. Let $\mathbf{L} = \mathbf{A}\mathbf{A}^T(\mathbf{A}\mathbf{A}^T)^+$ and $\mathbf{M} = \mathbf{A}\mathbf{A}^T$. We observe that the columns of $\mathbf{P}\mathbf{C}\mathbf{C}^T\mathbf{P}_{\backslash \ell}$ lie in the column span of $\mathbf{A}$. Therefore,

$$\mathrm{Tr}\left(\mathbf{Y}\mathbf{P}_\ell\mathbf{C}\mathbf{C}^T\mathbf{P}_{\backslash \ell}\mathbf{Y}\right) = \mathrm{Tr}\left(\mathbf{Y}\mathbf{L}\mathbf{P}_\ell\mathbf{C}\mathbf{C}^T\mathbf{P}_{\backslash \ell}\mathbf{Y}\right) \tag{A.35}$$

Then, by Cauchy-Schwarz,

$$\mathrm{Tr}\left(\mathbf{Y}\mathbf{L}\mathbf{P}_\ell\mathbf{C}\mathbf{C}^T\mathbf{P}_{\backslash \ell}\mathbf{Y}\right) \leq \sqrt{\mathrm{Tr}\left(\mathbf{Y}\mathbf{L}\mathbf{M}\mathbf{Y}\right)\mathrm{Tr}\left(\mathbf{P}_{\backslash \ell}\mathbf{C}\mathbf{C}^T\mathbf{P}_\ell\mathbf{M}^+\mathbf{P}_\ell\mathbf{C}\mathbf{C}^T\mathbf{P}_{\backslash \ell}\right)}$$

$$= \sqrt{\mathrm{Tr}\left(\mathbf{Y}\mathbf{M}\mathbf{Y}\right)\mathrm{Tr}\left(\mathbf{P}_{\backslash \ell}\mathbf{C}\mathbf{C}^T\mathbf{U}_\ell\Sigma^{-2}\mathbf{U}_\ell^T\mathbf{C}\mathbf{C}^T\mathbf{P}_{\backslash \ell}\right)} \tag{A.36}$$

$$= \sqrt{\mathrm{Tr}\left(\mathbf{Y}\mathbf{M}\mathbf{Y}\right)} \cdot \sqrt{\|\mathbf{P}_{\backslash \ell}\mathbf{C}\mathbf{C}^T\mathbf{U}_\ell\Sigma_\ell^{-1}\|_F^2}$$

Note, the first term is $\|\mathbf{A} - \mathbf{X}\mathbf{A}\|_F$. Therefore, we focus on the second term :

$$\|\mathbf{P}_{\backslash \ell}\mathbf{C}\mathbf{C}^T\mathbf{U}_\ell\Sigma_\ell^{-1}\|_F^2 = \sum_{i \in [\ell]} \|\mathbf{P}_{\backslash \ell}\mathbf{C}\mathbf{C}^T\mathbf{U}_{i,*}\|_2^2\sigma_i^{-2} \tag{A.37}$$

In order to bound the sum above, we bound each summand individually. Let $p_i$ be a unit vector in the direction of $\mathbf{C}\mathbf{C}^T\mathbf{U}_{i,*}$'s projection on $\mathbf{P}_{\backslash \ell}$. Then,

$$\|\mathbf{P}_{\backslash \ell}\mathbf{C}\mathbf{C}^T\mathbf{U}_{i,*}\|_2^2 = (p_i^T\mathbf{C}\mathbf{C}^T\mathbf{U}_{i,*})^2 \tag{A.38}$$

Let $\ell = \sigma_i^{-1}u_i + \frac{\sqrt{k}}{\|\mathbf{A}\|_F^2}p_i$. By Theorem A.1 we know that

$$\ell^T\mathbf{C}\mathbf{C}^T\ell - \frac{\epsilon\|A\|_F^2}{k}\ell^T\ell \leq \ell^T\mathbf{A}\mathbf{A}^T\ell \tag{A.39}$$

Substituting $\ell$ in the equation above,

$$\frac{\mathbf{U}_{i,*}\mathbf{C}\mathbf{C}^T\mathbf{U}_{i,*}}{\sigma_i^2} + \frac{kp_i^T\mathbf{C}\mathbf{C}^Tp_i}{\|\mathbf{A}\|_F^2} + \frac{2\sqrt{k}}{\|\mathbf{A}\|_F}p_i^T\mathbf{C}\mathbf{C}^T\mathbf{U}_{i,*} \leq \frac{\mathbf{U}_{i,*}\mathbf{M}\mathbf{U}_{i,*}}{\sigma_i^2} + \frac{k}{\|\mathbf{A}\|_F^2} + \frac{\epsilon\|\mathbf{A}\|_F^2}{k}\ell^T\ell$$

$$= 1 + \frac{k}{\|\mathbf{A}\|_F^2}p_i^T\mathbf{A}\mathbf{A}^Tp_i + \frac{\epsilon\|\mathbf{A}\|_F^2}{k}\ell^T\ell \tag{A.40}$$

Combining the above equation with (A.26) $\mathbf{U}_{i,*}^T\mathbf{C}\mathbf{C}^T\mathbf{U}_{i,*} \geq (1 - \epsilon)\mathbf{U}_{i,*}^T\mathbf{M}\mathbf{U}_{i,*} \geq (1 - \epsilon)\sigma_i^2$. Further, $p_i^T\mathbf{C}\mathbf{C}^Tp_i \geq p_i^T\mathbf{M}p_i - \frac{\epsilon\|\mathbf{A}\|_F^2}{k}$. Plugging this back into (A.40),

$$(1 - \epsilon)\left(\frac{\mathbf{U}_{i,*}\mathbf{M}\mathbf{U}_{i,*}}{\sigma_i^2} + \frac{kp_i^T\mathbf{M}p_i}{\|\mathbf{A}\|_F^2}\right) + \frac{2\sqrt{k}p_i\mathbf{C}\mathbf{C}^Tu_i}{\sigma_i\|\mathbf{A}\|_F}$$

$$\leq 1 + \frac{k}{\|\mathbf{A}\|_F^2}p_i^T\mathbf{M}p_i + \frac{\epsilon\|\mathbf{A}\|_F^2}{k}\ell^T\ell + 4\epsilon \tag{A.41}$$

Recall $p_i$ lies in the column space of $\mathbf{U}_{\setminus \ell}$, and thus $p_i \mathbf{M} p_i \leq \frac{\|A\|_F^2}{k}$ and thus we get

$$\frac{2\sqrt{k}}{\sigma_i \|\mathbf{A}\|_F^2} p_i^T \mathbf{C}\mathbf{C}^T u_i \leq 8\epsilon + \frac{\epsilon \|\mathbf{A}\|_F^2}{k} \ell^T \ell \leq 12\epsilon \tag{A.42}$$

Assuming again that $\epsilon < 1/2$ and observing that $\|\ell\|_2^2 \leq \frac{4k}{\epsilon \|\mathbf{A}\|_F^2}$ we get

$$\frac{2\sqrt{k} p_i \mathbf{C}\mathbf{C}^T u_i}{\sigma_i \|\mathbf{A}\|_F} \leq 12\epsilon$$
$$(p_i \mathbf{C}\mathbf{C}^T u_i^T)^2 \leq 144\epsilon^2 \frac{\sigma_i^2 \|\mathbf{A}\|_F^2}{k} \tag{A.43}$$

Plugging this back into $(A.40)$, we get that

$$\|\mathbf{P}_{\setminus \ell} \mathbf{C}\mathbf{C}^T \mathbf{U}_\ell \mathbf{\Sigma}_\ell^{-1}\|_F^2 \leq 288\epsilon^2 \|\mathbf{A}\|_F^2 \tag{A.44}$$

Since we have now bounded the second term, we plug it back into (A.36),

$$\mathrm{Tr}\left(\mathbf{Y}\mathbf{L}\mathbf{P}_\ell \mathbf{C}\mathbf{C}^T \mathbf{P}_{\setminus \ell} \mathbf{Y}\right) \leq \sqrt{\mathrm{Tr}\left(\mathbf{Y}\mathbf{M}\mathbf{Y}\right)} \sqrt{288 \|\mathbf{A}\|_F^2} \leq 17\epsilon \mathrm{Tr}\left(\mathbf{Y}\mathbf{A}\mathbf{A}^T \mathbf{Y}\right) \tag{A.45}$$

Combining $(A.24)$ $(A.27)$, $(A.34)$ and $(A.45)$, we get

$$\mathrm{Tr}\left(\mathbf{Y}\mathbf{C}\mathbf{C}^T \mathbf{Y}\right) = \mathrm{Tr}\left(\mathbf{Y}\mathbf{A}_\ell \mathbf{A}_\ell^T \mathbf{Y}\right) + \mathrm{Tr}\left(\mathbf{Y}\mathbf{A}_{\setminus \ell} \mathbf{A}_{\setminus \ell}^T \mathbf{Y}\right) \pm 40\epsilon \mathrm{Tr}\left(\mathbf{Y}\mathbf{A}\mathbf{A}^T \mathbf{Y}\right) \pm 10\epsilon \|\mathbf{A}\|_F^2 \tag{A.46}$$

Since $\mathrm{Tr}\left(\mathbf{Y}\mathbf{A}\mathbf{A}^T \mathbf{Y}\right) \leq \|\mathbf{A}\|_F^2$, rescaling $\epsilon$ by a constant finishes the proof. $\qquad\square$

We note that the critical ingredient in the proofs was estimating the column norms in sublinear time. We observe that we can also estimate the row norms in sublinear time and immediately obtain a Row Projection-Cost Preservation Theorem.

**Corollary A.2.** *(Row Projection-Cost Preservation.) Let $\mathbf{A}$ be a $m \times n$ matrix such that $\mathbf{A}$ satisfies approximate triangle inequality. For $i \in [n]$, let $\widetilde{X}_i$ be an $O\left(\frac{n}{b}\right)$-approximate estimate for the $i^{th}$ row of $\mathbf{A}$ such that it satisfies the guarantee of Lemma 2.1. Then, let $p = \{p_1, p_2 \ldots p_n\}$ be a probability distribution over the rows of $\mathbf{A}$ such that $p_i = \frac{\widetilde{X}_i}{\sum_{i'} \widetilde{X}_{i'}}$. Let $t = O\left(\frac{nk^2}{b\epsilon^2} \log(\frac{n}{\delta})\right)$. Then, construct $\mathbf{C}$ using $t$ rows of $\mathbf{A}$ and set each one to $\frac{A_{i,*}}{\sqrt{tp_i}}$ with probability $p_i$. With probability at least $1 - \delta$, for any rank-$k$ orthogonal projection $\mathbf{X}$*

$$\|\mathbf{C} - \mathbf{C}\mathbf{X}\|_F^2 = \|\mathbf{A} - \mathbf{A}\mathbf{X}\|_F^2 \pm \epsilon \|\mathbf{A}\|_F^2$$

Next, we describe how to apply projection-cost preserving sketching for low-rank approximation. Let $\mathbf{C}$ be a *column pcp* for $\mathbf{A}$. Then, an approximate solution for the best rank-$k$ approximation to $\mathbf{C}$ is an approximate solution for the best rank-$k$ approximation to $\mathbf{A}$. Formally,

**Lemma A.1.** *Let $\mathbf{C}$ be a column pcp for $\mathbf{A}$ satisfying the guarantee of Theorem A.2. Let $\mathbf{P}_\mathbf{C}^*$ be the projection matrix that minimizes $\|\mathbf{C} - \mathbf{X}\mathbf{C}\|_F^2$ and $\mathbf{P}_\mathbf{A}^*$ be the projection matrix that minimizes $\|\mathbf{A} - \mathbf{X}\mathbf{A}\|_F^2$. Then, for any projection matrix $\mathbf{P}$ such that $\|\mathbf{C} - \mathbf{P}\mathbf{C}\|_F^2 \leq \|\mathbf{C} - \mathbf{P}_\mathbf{C}^*\mathbf{C}\|_F^2 + \epsilon \|\mathbf{C}\|_F^2$, with probability at least $98/100$,*

$$\|\mathbf{A} - \mathbf{P}\mathbf{A}\|_F^2 \leq \|\mathbf{A} - \mathbf{P}_\mathbf{A}^*\mathbf{A}\|_F^2 + \epsilon \|\mathbf{A}\|_F^2$$

*A similar guarantee holds if $\mathbf{C}$ is a row pcp of $\mathbf{A}$.*

*Proof.* By the optimality of $\mathbf{P}_C^*$, we know that $\|\mathbf{C} - \mathbf{P}\mathbf{C}\|_F^2 \leq \|\mathbf{C} - \mathbf{P}_\mathbf{C}^*\mathbf{C}\|_F^2 + \epsilon \|\mathbf{C}\|_F^2 \leq \|\mathbf{C} - \mathbf{P}_\mathbf{A}^*\mathbf{C}\|_F^2 + \epsilon \|\mathbf{C}\|_F^2$. Since $\mathbf{C}$ is a *column pcp* of $\mathbf{A}$, $\|\mathbf{C} - \mathbf{P}_\mathbf{A}^*\mathbf{C}\|_F^2 \leq \|\mathbf{A} - \mathbf{P}_\mathbf{A}^*\mathbf{A}\|_F^2 + \epsilon \|\mathbf{A}\|_F^2$, therefore, with probability at least $99/100$,

$$\|\mathbf{C} - \mathbf{P}\mathbf{C}\|_F^2 \leq \|\mathbf{A} - \mathbf{P}_\mathbf{A}^*\mathbf{A}\|_F^2 + \epsilon \|\mathbf{A}\|_F^2 + \epsilon \|\mathbf{C}\|_F^2 \leq \|\mathbf{A} - \mathbf{P}_\mathbf{A}^*\mathbf{A}\|_F^2 + O(\epsilon) \|\mathbf{A}\|_F^2 \tag{A.47}$$

where the last inequality follows from $\mathbb{E}[\mathbf{C}] = \mathbf{A}$ and Markov's bound. Similarly, $\|\mathbf{C} - \mathbf{P}\mathbf{C}\|_F^2 \geq \|\mathbf{A} - \mathbf{P}\mathbf{A}\|_F^2 - \epsilon \|\mathbf{A}\|_F^2$, therefore, with probability at least $99/100$,

$$\|\mathbf{C} - \mathbf{P}\mathbf{C}\|_F^2 \geq \|\mathbf{A} - \mathbf{P}\mathbf{A}\|_F^2 - O(\epsilon) \|\mathbf{A}\|_F^2 \tag{A.48}$$

Union bounding over the two events and combining the two equations, with probability at least $98/100$ we get

$$\|\mathbf{A} - \mathbf{P}\mathbf{A}\|_F^2 \leq \|\mathbf{A} - \mathbf{P}_\mathbf{A}^*\mathbf{A}\|_F^2 + 4\epsilon\|\mathbf{A}\|_F^2 \tag{A.49}$$

Rescaling $\epsilon$ completes the proof. We note that a similar lemma holds if $\mathbf{C}$ is a *row pcp* of $\mathbf{A}$. $\square$

**Lemma A.1.** *Let $\mathbf{C}$ be a column pcp for $\mathbf{A}$ satisfying the guarantee of Theorem A.2. Let $\mathbf{P}_\mathbf{C}^*$ be the projection matrix that minimizes $\|\mathbf{C} - \mathbf{X}\mathbf{C}\|_F^2$ and $\mathbf{P}_\mathbf{A}^*$ be the projection matrix that minimizes $\|\mathbf{A} - \mathbf{X}\mathbf{A}\|_F^2$. Then, for any projection matrix $\mathbf{P}$ such that $\|\mathbf{C} - \mathbf{P}\mathbf{C}\|_F^2 \leq \|\mathbf{C} - \mathbf{P}_\mathbf{C}^*\mathbf{C}\|_F^2 + \epsilon\|\mathbf{C}\|_F^2$, with probability at least $98/100$,*

$$\|\mathbf{A} - \mathbf{P}\mathbf{A}\|_F^2 \leq \|\mathbf{A} - \mathbf{P}_\mathbf{A}^*\mathbf{A}\|_F^2 + \epsilon\|\mathbf{A}\|_F^2$$

*A similar guarantee holds if $\mathbf{C}$ is a row pcp of $\mathbf{A}$.*

*Proof.* By the optimality of $\mathbf{P}_C^*$, we know that $\|\mathbf{C} - \mathbf{P}\mathbf{C}\|_F^2 \leq \|\mathbf{C} - \mathbf{P}_\mathbf{C}^*\mathbf{C}\|_F^2 + \epsilon\|\mathbf{C}\|_F^2 \leq \|\mathbf{C} - \mathbf{P}_\mathbf{A}^*\mathbf{C}\|_F^2 + \epsilon\|\mathbf{C}\|_F^2$. Since $\mathbf{C}$ is a *column pcp* of $\mathbf{A}$, $\|\mathbf{C} - \mathbf{P}_\mathbf{A}^*\mathbf{C}\|_F^2 \leq \|\mathbf{A} - \mathbf{P}_\mathbf{A}^*\mathbf{A}\|_F^2 + \epsilon\|\mathbf{A}\|_F^2$, therefore, with probability at least $99/100$,

$$\|\mathbf{C} - \mathbf{P}\mathbf{C}\|_F^2 \leq \|\mathbf{A} - \mathbf{P}_\mathbf{A}^*\mathbf{A}\|_F^2 + \epsilon\|\mathbf{A}\|_F^2 + \epsilon\|\mathbf{C}\|_F^2 \leq \|\mathbf{A} - \mathbf{P}_\mathbf{A}^*\mathbf{A}\|_F^2 + O(\epsilon)\|\mathbf{A}\|_F^2$$

where the last inequality follows from $\mathbb{E}[\mathbf{C}] = \mathbf{A}$ and Markov's bound. Similarly, $\|\mathbf{C} - \mathbf{P}\mathbf{C}\|_F^2 \geq \|\mathbf{A} - \mathbf{P}\mathbf{A}\|_F^2 - \epsilon\|\mathbf{A}\|_F^2$, therefore, with probability at least $99/100$,

$$\|\mathbf{C} - \mathbf{P}\mathbf{C}\|_F^2 \geq \|\mathbf{A} - \mathbf{P}\mathbf{A}\|_F^2 - O(\epsilon)\|\mathbf{A}\|_F^2$$

Union bounding over the two events and combining the two equations, with probability at least $98/100$ we get

$$\|\mathbf{A} - \mathbf{P}\mathbf{A}\|_F^2 \leq \|\mathbf{A} - \mathbf{P}_\mathbf{A}^*\mathbf{A}\|_F^2 + 4\epsilon\|\mathbf{A}\|_F^2$$

Rescaling $\epsilon$ completes the proof. We note that a similar lemma holds if $\mathbf{C}$ is a *row pcp* of $\mathbf{A}$. $\square$

**Lemma A.2.** *(Scalar Chernoff Bound.) Let $\mathbf{A} \in \mathbb{R}^{m \times n}$ satisfy approximate triangle inequality. For $j \in [n]$, let $\widetilde{X}_j$ be a $O\left(\frac{m}{b}\right)$-approximate estimate for the $j^{th}$ column of $A$ such that it satisfies the guarantee of Corollary A.1. Then, let $q = \{q_1, q_2 \ldots q_n\}$ be a probability distribution over the columns of $A$ such that $q_j = \frac{\widetilde{X}_j}{\sum_{j'} \widetilde{X}_{j'}}$. Let $t = O\left(\frac{mk^2}{b\epsilon^2}\log(\frac{m}{\delta})\right)$ for some constant c. Construct $\mathbf{C}$ using $t$ columns of $A$ and set each one to $\frac{A_{*,j}}{\sqrt{tq_j}}$ with probability $q_j$. Let $\ell$ be the index of the smallest singular value of $\mathbf{A}$ such that $\sigma_\ell^2 \geq \frac{\|\mathbf{A}\|_F^2}{k}$. With probability at least $1 - \delta$,*

$$\mathrm{Tr}\left(\mathbf{A}_{\backslash \ell}\mathbf{A}_{\backslash \ell}^T\right) - \mathrm{Tr}\left(\mathbf{P}_{\backslash \ell}\mathbf{C}\mathbf{C}^T\mathbf{P}_{\backslash \ell}\right) = \pm\epsilon\|\mathbf{A}\|_F^2 \tag{A.50}$$

*Proof.* We can rewrite the above equation as $\|\mathbf{P}_{\backslash \ell}\mathbf{C}\|_F^2 - \|\mathbf{A}_{\backslash \ell}\|_F^2 = \pm\epsilon\|\|\|_F^2$. By summing over the column norms of $\mathbf{C}$, we get $\|\mathbf{P}_{\backslash \ell}\mathbf{C}\|_F^2 = \sum_{j=[t]} \|\mathbf{P}_{\backslash \ell}\mathbf{C}_{*,j}\|_2^2$. Next, we upper bound each term in the sum as follows :

$$\|\mathbf{P}_{\backslash \ell}\mathbf{C}_{*,j}\|_2^2 = \frac{1}{tq_j}\|\mathbf{P}_{\backslash \ell}\mathbf{A}_{*,j}\|_2^2$$

$$= \left(\frac{b\epsilon^2}{mk^2\log(m/\delta)}\right)\left(\frac{m\|\mathbf{A}\|_F^2}{b\|\mathbf{A}_{*,j}\|_2^2}\right)\|\mathbf{P}_{\backslash \ell}\mathbf{A}_{*,j}\|_2^2 \tag{A.51}$$

$$\leq \left(\frac{\epsilon^2}{k^2\log(m/\delta)}\right)\|\mathbf{A}\|_F^2$$

Note, $\frac{k^2\log(m/\delta)}{\epsilon^2\|\mathbf{A}\|_F^2}\|\mathbf{P}_{\backslash \ell}\mathbf{C}_{*,j}\|_2^2 \in [0,1]$ and $\mathbb{E}\left[\sum_{j=[t]}\|\mathbf{P}_{\backslash \ell}\mathbf{C}_{*,j}\|_2^2\right] = \|\mathbf{A}_{\backslash \ell}\|_2^2$. By Chernoff,

$$\mathbf{Pr}\left[\|\mathbf{P}_{\backslash \ell}\mathbf{C}\|_F^2 \geq \|\mathbf{P}_{\backslash \ell}\mathbf{A}\|_F^2 + \epsilon\|\mathbf{A}\|_F\right]$$

$$= \mathbf{Pr}\left[\left(\frac{k^2\log(m/\delta)}{\epsilon^2\|\mathbf{A}\|_F^2}\right)\sum_{j\in[t]}\|\mathbf{P}_{\backslash \ell}\mathbf{C}_{*,j}\|_F^2 \geq \left(\frac{k^2\log(m/\delta)}{\epsilon^2\|\mathbf{A}\|_F^2}\right)\|\mathbf{A}_{\backslash \ell}\|_F^2 + \epsilon\|\mathbf{A}\|_F\right]$$

$$= \mathbf{Pr}\left[\left(\frac{k^2\log(m/\delta)}{\epsilon^2\|\mathbf{A}\|_F^2}\right)\sum_{j\in[t]}\|\mathbf{P}_{\backslash \ell}\mathbf{C}_{*,j}\|_F^2 \geq 1 + \frac{\epsilon\|\mathbf{A}\|_F}{\|\mathbf{A}_{\backslash \ell}\|_F^2}\left(\frac{k^2\log(m/\delta)\|\mathbf{A}_{\backslash \ell}\|_F^2}{\epsilon^2\|\mathbf{A}\|_F^2}\right)\right] \tag{A.52}$$

$$\leq e^{-\frac{c\log(m/\delta)}{4}} \leq \delta/2$$

Therefore, with probability at least $1 - \delta/2$, $\|\mathbf{P}_{\backslash \ell}\mathbf{C}\|_F^2 - \|\mathbf{A}_{\backslash \ell}\|_F^2 \leq \epsilon\|\mathbf{A}\|_F$. Similarly, we can show that with probability at least $1 - \delta/2$, $\|\mathbf{P}_{\backslash \ell}\mathbf{C}\|_F^2 - \|\mathbf{A}_{\backslash \ell}\|_F^2 \geq -\epsilon\|\mathbf{A}\|_F$. Union bounding over the two events finishes the proof. $\qquad\square$

### A.6  Full Sublinear Time Algorithm

---

**Algorithm 3 : Full Sublinear Time Algorithm.**

---

**Input:** A Distance Matrix $\mathbf{A}_{m \times n}$, integer $k$, $\epsilon > 0$ and a small constant $\gamma > 0$.

1. Let $2r = O\left(1/\gamma\right)$ and let $\mathbf{A}_{(0)} = \mathbf{A}$, $s_0 = n$ and $t_0 = m$. Set $b_1 = m^\gamma$, $b_2 = n^\gamma$.

2. For $i \in [2r]$, recursively construct matrix $\mathbf{A}_{(i)}$ as follows:

   (a) If $i$ is odd, run $\texttt{ColumnNormEstimation}(\mathbf{A}_{(i-1)}, b_2)$. By Lemma A.3, we obtain $O\left(\frac{t_{i-1}}{b_2}\right)$-approximate estimates of the column norms of $\mathbf{A}_{(i-1)}$. Let $q = \{q_1, q_2 \ldots q_{s_{i-1}}\}$ denote a distribution over columns of $\mathbf{A}_{(i-1)}$ proportional to the relative estimate for each column.
   Construct a *column pcp* for $\mathbf{A}_{(i-1)}$ by sampling $s_i = \widetilde{\Theta}\left(\frac{s_{i-1}}{b_2}\text{poly}(\frac{k}{\epsilon})\right)$ columns such that each column is set to $\frac{(\mathbf{A}_{(i-1)})_{*,j}}{\sqrt{s_i q_j}}$ with probability $q_j$. Let $\mathbf{A}_{(i)} = \mathbf{A}_{(i-1)}\mathbf{S}_i$ be the resulting $t_{i-1} \times s_i$ matrix that follows guarantees of Theorem A.2.

   (b) If $i$ is even, run $\texttt{RowNormEstimation}(\mathbf{A}_{(i-1)}, b_1)$. By Lemma A.3, we obtain $O\left(\frac{s_{-1}}{b_1}\right)$-approximate estimates of the row norms of $\mathbf{A}_{(i-1)}$. Let $p = \{p_1, p_2 \ldots p_{t_{i-1}}\}$ denote a distribution over rows of $\mathbf{A}_{(i-1)}$ proportional to the relative estimate for each row.
   Construct a *row pcp* for $\mathbf{A}_{(i-1)}$ by sampling $t_i = \widetilde{\Theta}\left(\frac{t_{i-1}}{b_1}\text{poly}(\frac{k}{\epsilon})\right)$ columns such that each column is set to $\frac{(\mathbf{A}_{(i-1)})_{\ell,*}}{\sqrt{t_i p_\ell}}$ with probability $p_\ell$. Let $\mathbf{A}_{(i)} = \mathbf{T}_i\mathbf{A}_{(i-1)}$ be the resulting $t_i \times s_{i-1}$ matrix that follows guarantees of Corollary A.2.

3. Let $\mathbf{A}_{(2r)}$ be the final matrix at the end of the recursion. Compute the truncated SVD $(\mathbf{A}_{(2r)}, k) = \mathbf{U}_{2r}\Sigma_{2r}\mathbf{V}_{2r}^T$. Let $\mathbf{V}_{2r}^T$ represent the top $k$ singular vectors in the row space. Construct a leverage-score sketching matrix $\mathbf{E}_{2r}$ with $\text{poly}(\frac{k}{\epsilon})$ columns using the leverage scores of $\mathbf{V}_{2r}^T$, following the guarantees of Theorem 4.3. Compute $\mathbf{X}_{\mathbf{A}_{(2r-1)}} = \text{argmin}_{\mathbf{X}}\|\mathbf{A}_{(2r-1)}\mathbf{E}_{2r-1} - \mathbf{X}\mathbf{V}_{2r}^T\mathbf{E}_{2r-1}\|$.

4. Compute a decomposition $\mathbf{U}_{2r-1}\mathbf{V}_{2r-1}$ of $\mathbf{X}_{\mathbf{A}_{(2r-1)}}\mathbf{V}_{2r}^T$ such that $\mathbf{U}_{2r-1}$ has orthonormal columns. Consider the regression problem $\min_{\mathbf{X}}\|\mathbf{A}_{(2r-2)} - \mathbf{U}_{2r-1}\mathbf{X}\|_F^2$. Compute $\mathbf{X}_{\mathbf{A}_{(2r-2)}} = \text{argmin}_{\mathbf{X}}\|\mathbf{E}_{2r-2}\mathbf{A}_{(2r-2)} - \mathbf{E}_{2r-2}\mathbf{U}_{2r-1}\mathbf{X}\|_F^2$, where $\mathbf{E}_{2r-2}$ is a leverage score sketching matrix with $\text{poly}(\frac{k}{\epsilon})$ rows constructed according to Theorem 4.3.

5. Let $\mathbf{U}_{2r-1}\mathbf{X}_{\mathbf{A}_{(2r-2)}}$ be the starting point for $\mathbf{A}_{2r-3}$ as $\mathbf{U}_{2r}\Sigma_{2r}\mathbf{V}_{2r}^T$ was for $\mathbf{A}_{2r-1}$ and recurse to the top. Let $\mathbf{U}_1, \mathbf{X}_{\mathbf{A}}$ be the solution obtain from solving $\min_{\mathbf{X}}\|\mathbf{E}\mathbf{A} - \mathbf{E}\mathbf{U}_1\mathbf{X}\|_F^2$, where $\mathbf{E}$ is a leverage score sketching matrix with $\text{poly}(\frac{k}{\epsilon})$ rows, constructed according to Theorem 4.3.

**Output:** $\mathbf{M} = \mathbf{U}_1$, $\mathbf{N}^T = \mathbf{X}_{\mathbf{A}}$

---

In this section we improve the running time of our previous sublinear time algorithm. Intuitively, our algorithm recursively constructs projection-cost preserving sketches for the rows and columns of the original matrix by sampling according to coarse estimates of the row and column norms. Note, we are able to obtain these estimates by dividing the subsampled matrices at each step into weight classes such that each weight class approximately satisfies triangle inequality. At the bottom of the recursion we reduce the input matrix $\mathbf{A}$ to a $\text{poly}(\frac{k}{\epsilon}) \times \text{poly}(\frac{k}{\epsilon})$ matrix, for which we can compute the SVD in $O(\text{poly}(\frac{k}{\epsilon}))$ time.

Starting with an orthonormal basis of the SVD, we alternate between approximately computing the best rank-$k$ projection in the column and the row space all the way up the recursion chain and output the final rank-$k$ matrix. However, computing the SVD or even running an input-sparsity time algorithm near the top becomes prohibitively expensive and is no longer sublinear. Therefore, we find approximate solutions to the best rank-$k$ column and row subspaces by formulating a regression problem, sketching it to a smaller dimension using leverage score sampling and solving it approximately.

We show that recursive sampling indeed approximately preserves rank-$k$ subspaces of the row and column space. For the sake of brevity throughout the rest of the analysis, let $\mathbf{A}_{(i)}$ be a $t_i \times s_i$ matrix created by recursively sampling rows or columns of $\mathbf{A}_{(i-1)}$ such that at each step the *row* or *column pcp* properties are satisfied. Formally, let $\mathbf{A}_{(0)} = \mathbf{A}$ be a $t_0 \times s_0$ matrix, where $t_0 = m$ and $s_0 = n$. Then, if $i$ is odd, $\mathbf{A}_{(i)} = \mathbf{A}_{(i-1)}\mathbf{S}_i$ is a $t_{i-1} \times s_i$ matrix and a *column pcp* for $\mathbf{A}_{(i-1)}$ and if $i$ is even, $\mathbf{A}_{(i)} = \mathbf{T}_i\mathbf{A}_{(i-1)}$ is a $t_i \times s_{i-1}$ matrix and a *row pcp* for $\mathbf{A}_{(i-1)}$. We note that $s_i = \widetilde{\Theta}\left(\frac{s_{i-1}}{b_2}\mathrm{poly}(\frac{k}{\epsilon})\right)$ and $t_i = \widetilde{\Theta}\left(\frac{t_{i-1}}{b_1}\mathrm{poly}(\frac{k}{\epsilon})\right)$.

To address the issue of rescaling every time we subsample rows or columns, we split the rows or columns of $\mathbf{A}_{(i)}$ into $O(\epsilon^{-1}\log(mn))$ weight classes such that triangle inequality approximately holds in each weight class. Therefore, the column or row norm estimation algorithm goes through for each weight class independently and we obtain an overall $O(\frac{m}{b_2})$-approximation to the column norms at the cost of reading $O(\epsilon^{-1}\log(m)b_2)$ entries per column. A similar guarantee holds for the row norms. This idea simply extends to the recursive algorithm as we can create weight classes at each recursive step run the simple row and column norm estimation algorithms.

Intuitively, to handle the first column rescaling , we partition the columns of the matrix into $O(\epsilon^{-1}\log(mn))$ blocks such that each block satisfies approximate triangle inequality. By Lemma 2.1, we can estimate the row norms for each block efficiently, and summing the estimates suffices to obtain an approximation to the row norms. Next, we subsample the rows and scale them. However, we observe that we can yet again partition each sub-matrix that satisfies approximate triangle inequality into $O(\epsilon^{-1}\log(mn))$ weight classes and yet again satisfy approximate triangle inequality. We note that we only recurse a constant $(1/\gamma)$ number of times, therefore the total number of sub-matrices formed is $O\left(\left(\epsilon^{-1}\log(mn)\right)^{\frac{1}{\gamma}}\right) = \mathrm{poly}(\epsilon^{-1}\log(mn))$ and the run-time blows up by at most that factor.

**Lemma A.3.** *(Estimating row and column norms under rescaling.) Let $\mathbf{A}$ be a $m \times n$ matrix such that $\mathbf{A}$ satisfies approximate triangle inequality. For a small fixed constant $\gamma$, and for all $i \in [1/\gamma]$, let $\mathbf{A}_{(i)}$ be a $t_i \times s_i$ scaled sub-matrix of $\mathbf{A}$ as defined as above. There exists an algorithm that, with probability at least $9/10$, obtains a $O\left(\frac{s_{i-1}}{b_1}\right)$-approximation to the row norms of $\mathbf{A}_{(i)}$ in $\widetilde{O}(b_2 m + n)$ time. A similar guarantee holds for estimating column norms.*

*Proof.* We begin with a $m \times n$ matrix $\mathbf{A}$ that satisfies approximate triangle inequality. It follows from Corollary A.1 that we can approximately estimate column norms of $\mathbf{A}$ in $O(b_2 m + n)$ time. Therefore, we can construct the *column pcp*, $\mathbf{A}_{(1)} = \mathbf{A}\mathbf{S}_1$, such that the $j^{th}$ column of $\mathbf{A}$, if sampled, is rescaled by $\frac{1}{\sqrt{s_1 q_j}}$. Therefore, the resulting matrix, $\mathbf{A}_{(1)}$ may no longer be a distance matrix. As discussed in the previous section, to address this issue, we partition the columns of $\mathbf{A}_{(1)}$ into weight classes such that the $g^{th}$ weight class contains column index $j$ if the corresponding scaling factor $\frac{1}{\sqrt{q_j}}$ lies in the interval $\left[(1+\epsilon)^g, (1+\epsilon)^{g+1}\right)$. Note, we can ignore the $(\frac{1}{\sqrt{s_1}})$-factor since every entry is rescaled by the same constant. Formally,

$$\mathcal{W}_g^{(1)} = \left\{i \in [s_1] \,\Big|\, \frac{1}{\sqrt{q_j}} \in \left[(1+\epsilon)^g, (1+\epsilon)^{g+1}\right)\right\} \tag{A.53}$$

Next, with high probability, for all $j \in [n]$, if column $j$ is sampled, $\frac{1}{q_j} \le n^c$ for a large constant $c$. If instead, $q_j \le \frac{1}{n^{c'}}$, the probability that the $j^{th}$ is sampled would be at most $1/n^{c'}$, for some $c' > c$. Union bounding over such events for $n$ columns, the number of weight classes is at most $\log_{1+\epsilon}(n^c) = O\left(\epsilon^{-1}\log(n)\right)$. Let $\mathbf{A}_{(1)|\mathcal{W}_g^{(1)}}$ denote the columns of $\mathbf{A}_{(1)}$ restricted to the set of indices in $\mathcal{W}_g$. Observe that all entries in $\mathbf{A}_{(1)|\mathcal{W}_g^{(1)}}$ are scaled to within a $(1+\epsilon)$-factor of each

other and therefore, satisfy approximate triangle inequality (equation 2.1). Therefore, row norms of $\mathbf{A}_{(1)|\mathcal{W}_g^{(1)}}$ can be computed using Algorithm 1 and the estimator is an $O\left(\frac{n}{b_2}\right)$-approximation (for some parameter $b_2$), since Lemma 2.1 blows up by a factor of at most $1 + \epsilon$. Summing over the estimates from each partition above, with probability at least 99/100, we obtain an $O\left(\frac{n}{b_2}\right)$-approximate estimate to row norms of $\mathbf{A}_{(1)}$. However, we note that each iteration of Algorithm 1 reads $b_2 m + n$ entries of $\mathbf{A}$ and there are at most $O(\epsilon^{-1} \log(n))$ iterations. Therefore, the time taken to compute the estimates to the row norms is $O\left((b_2 m + n)\epsilon^{-1} \log(n)\right)$.

Now, we can construct a *row pcp* of $\mathbf{A}_{(1)}$, which we denote by $\mathbf{A}_{(2)}$, such that each row in $\mathbf{A}_{(2)}$ is a scaled subset of the rows of $\mathbf{A}_{(1)}$. Our next task is to estimate the column norms of $\mathbf{A}_{(2)}$. It suffices to show that $\mathbf{A}_{(2)}$ can be partitioned into a small number of sub-matrices such that each one satisfies $(1 + \epsilon)$-approximate triangle inequality.

Observe, we previously split the matrix $\mathbf{A}_{(1)}$ according to $\mathcal{W}_g^{(1)}$, into $O\left(\epsilon^{-1} \log(n)\right)$ sub-matrices such that each matrix satisfies $(1 + \epsilon)$-approximate triangle inequality. Consider one such sub-matrix, $\mathbf{A}_{(1)|\mathcal{W}_g^{(1)}}$. In the construction of the *row pcp* $\mathbf{A}_{(2)}$, we rescale a subset of rows of each of $\mathbf{A}_{(1)|\mathcal{W}_g^{(1)}}$. Therefore, we can again create $O\left(\epsilon^{-1} \log(m)\right)$ geometrically increasing weight classes for the rows of $\mathbf{A}_{(1)|\mathcal{W}_g^{(1)}}$. Note, restricting rows of $\mathbf{A}_{(1)|\mathcal{W}_g^{(1)}}$ to one weight class results in a sub-matrix that satisfies $(1 + \epsilon)^2$-approximate triangle inequality. We repeat the above analysis for each such sub-matrix, since we again start with a matrix that satisfies $(1 + \epsilon)$-approximate triangle inequality, after rescaling $\epsilon$ by a constant.

Critically, we note that we only repeat the recursion a constant number of times, therefore the approximation factor for triangle inequality blows up by $(1 + \epsilon)^{1/\gamma}$. Since $\gamma$ is a constant, we can rescale $\epsilon$ by a constant, and therefore, all sub-matrices satisfy $(1 + \epsilon)$-approximate triangle inequality. the total number of sub-matrices formed is $O\left(\left(\epsilon^{-1} \log(n) \log(m)\right)^{\frac{1}{\gamma}}\right) = \text{poly}(\epsilon^{-1} \log(n) \log(m))$ and the run-time blows up by at most that factor. $\qquad\square$

Note, we can now black-box the algorithm for estimating row and column norms of scaled sub-matrices of $\mathbf{A}$. For the sake of brevity, we do not include them in Algorithm 3. Next, we present a critical structural result that enables us to recursively apply the *pcp* guarantees.

**Lemma A.4.** *(Recursive PCP Lemma.) Let $\mathbf{A}_{(i)}$ be defined as above. Then, for any $\epsilon > 0$, integer $k$ and a small constant $\gamma > 0$, $2r = O\left(\frac{1}{\gamma}\right)$, simultaneously for all odd $i \in [2r]$, for all rank-$k$ projection matrices $\mathbf{X}_i$, with probability at least 98/100,*

$$\|\mathbf{A}_{(i)}(\mathbb{I} - \mathbf{X}_i)\|_F^2 = \|\mathbf{A}_{(i-1)}(\mathbb{I} - \mathbf{X}_i)\|_F^2 \pm \epsilon\|\mathbf{A}_{(i-1)}\|_F^2$$

*Further, let $\mathbf{X}^*_{\mathbf{A}_{(i)}}$ be the projection that minimizes $\|\mathbf{A}_{(i)}(\mathbb{I} - \mathbf{X}_i)\|_F^2$ and let $\mathbf{X}^*_{\mathbf{A}_{(i-1)}}$ be the projection that minimizes $\|\mathbf{A}_{(i-1)}(\mathbb{I} - \mathbf{X}_i)\|_F^2$. Then, simultaneously for all odd $i$, for any rank-$k$ projection matrix $\mathbf{X}_i$ such that $\|\mathbf{A}_{(i)}(\mathbb{I} - \mathbf{X}_i)\|_F^2 \leq \|\mathbf{A}_{(i)}(\mathbb{I} - \mathbf{X}^*_{\mathbf{A}_{(i)}})\|_F^2 + \epsilon\|\mathbf{A}_{(i)}\|_F^2$, with probability at least 98/100,*

$$\|\mathbf{A}_{(i-1)}(\mathbb{I} - \mathbf{X}_i)\|_F^2 \leq \|\mathbf{A}_{(i-1)}(\mathbb{I} - \mathbf{X}^*_{\mathbf{A}_{(i-1)}})\|_F^2 + \epsilon\|\mathbf{A}_{(i-1)}\|_F^2$$

*A similar guarantee holds if $i$ is even.*

*Proof.* For a given matrix $\mathbf{A}_{(i)}$, such that $i$ is odd, we employ Theorem A.2 with $\delta = 1/n^c$ for a fixed constant $c$. Note, the number of columns we sample only blows up by a constant and

$$\|\mathbf{A}_{(i)}(\mathbb{I} - \mathbf{X}_i)\|_F^2 \leq \|\mathbf{A}_{(i-1)}(\mathbb{I} - \mathbf{X}_i)\|_F^2 + \epsilon\|\mathbf{A}_{(i-1)}\|_F^2$$

holds with probability at least $1 - \frac{1}{n^c}$. Union bounding over all such events for $i \in [2r]$, such that $i$ is odd, the above guarantee holds simultaneously with probability at least $1 - 1/n^{c-1}$. Setting $\delta = 1/n^c$ in Corollary A.2, it follows that simultaneously for all $i \in [2r]$, such that $i$ is even,

$$\|(\mathbb{I} - \mathbf{X}_i)\mathbf{A}_{(i)}\|_F^2 \leq \|(\mathbb{I} - \mathbf{X}_i)\mathbf{A}_{(i-1)}\|_F^2 + \epsilon\|\mathbf{A}_{(i-1)}\|_F^2$$

with probability at least $1 - 1/n^{c-1}$. Note, for a fixed odd $i$, following the analysis of Lemma A.1,

$$\|\mathbf{A}_{(i-1)}(\mathbb{I} - \mathbf{X}_i)\|_F^2 \leq \|\mathbf{A}_{(i-1)}(\mathbb{I} - \mathbf{X}^*_{\mathbf{A}_{(i-1)}})\|_F^2 + O(\epsilon)\|\mathbf{A}_{(i-1)}\|_F^2$$

holds with probability at least $1 - \frac{1}{200r}$. Union bounding over all such events $i \in [2r]$, such that i is odd, the above guarantee holds simultaneously, with probability at least $99/100$.

Similarly, for even $i$, let $\mathbf{X}^*_{\mathbf{A}_{(i)}}$ be the projection that minimizes $\|(\mathbb{I} - \mathbf{X}_i)\mathbf{A}_{(i)}\|_F^2$ and let $\mathbf{X}^*_{\mathbf{A}_{(i-1)}}$ be the projection that minimizes $\|(\mathbb{I} - \mathbf{X}_i)\mathbf{A}_{(i-1)}\|_F^2$. Then, simultaneously for all even $i$, for any rank-$k$ projection matrix $\mathbf{X}_i$ such that

$$\|(\mathbb{I} - \mathbf{X}_i)\mathbf{A}_{(i)}\|_F^2 \leq \|(\mathbb{I} - \mathbf{X}^*_{\mathbf{A}_{(i)}})\mathbf{A}_{(i)}\|_F^2 + O(\epsilon)\|\mathbf{A}_{(i)}\|_F^2$$

with probability at least $99/100$, it holds that

$$\|(\mathbb{I} - \mathbf{X}_i)\mathbf{A}_{(i-1)}\|_F^2 \leq \|(\mathbb{I} - \mathbf{X}^*_{\mathbf{A}_{(i-1)}})\mathbf{A}_{(i-1)}\|_F^2 + O(\epsilon)\|\mathbf{A}_{(i-1)}\|_F^2$$

Union bounding over the odd and even events completes the proof. □

Using the above structural guarantees, we show that Algorithm 3 indeed achieves an additive error guarantee for low-rank approximation. Intuitively, at each recursive step we either approximately preserve all rank-$k$ row or column projections. We finally obtain a matrix that is independent of $m$ and $n$ and therefore we can compute its SVD. We then begin with the rank-$k$ matrix output by the SVD and critically rely on Lemma A.1 to switch between finding approximately optimal projections in the row and column space while climb back up the recursive stack.

**Lemma A.5.** *(Additive Error Guarantee.) Let $\mathbf{A} \in \mathbb{R}^{m \times n}$ be a matrix such that it satisfies approximate triangle inequality. Then, for any $\epsilon > 0$, integer $k$, and a small constant $\gamma > 0$, with probability at least $9/10$, Algorithm 3 outputs a rank-k matrix $\mathbf{MN}^T$ such that $\mathbf{M} \in \mathbf{R}^{m \times k}$, $\mathbf{N} \in \mathbf{R}^{n \times k}$ and*

$$\|\mathbf{A} - \mathbf{MN}^T\|_F^2 \leq \|\mathbf{A} - \mathbf{A}_k\|_F^2 + O(\epsilon)\|\mathbf{A}\|_F^2$$

*Proof.* By Lemma A.4, we know that approximately optimal rank-$k$ projection matrix for $\mathbf{A}_{(i)}$ is an approximately optimal rank-$k$ matrix for $\mathbf{A}_{(i-1)}$ up to an additive error term of $\epsilon\|\mathbf{A}_{(i-1)}\|_F^2$. Let $2r = O\left(\frac{1}{\gamma}\right)$ be the number of recursive calls. Concretely, since $\mathbf{A}_{(2r)}$ is a *row pcp* of $\mathbf{A}_{(2r-1)}$, we know that for all rank-$k$ projections $\mathbf{X}_{2r}$, with probability at least $98/100$,

$$\|\mathbf{A}_{(2r)}(\mathbb{I} - \mathbf{X}_{2r})\|_F^2 \leq \|\mathbf{A}_{(2r-1)}(\mathbb{I} - \mathbf{P}_{2r})\|_F^2 + \epsilon\|\mathbf{A}_{(2r-1)}\|_F^2 \tag{A.54}$$

Since $\mathbf{A}_{(2r)}$ is a small matrix of dimension, we can afford to compute $\mathrm{SVD}(\mathbf{A}_{(2r)})$ (we analyze this runtime below). Let $\mathbf{U}_{2r}\mathbf{D}_{2r}\mathbf{V}_{2r}^T$ be the truncated SVD, containing the top $k$ singular values and setting the rest to 0. Thus, we know that $\mathbf{V}_{2r}\mathbf{V}_{2r}^T$ is the optimal projection matrix for $\mathbf{A}_{(2r)}$, and by Lemma A.4,

$$\|\mathbf{A}_{(2r-1)}(\mathbb{I} - \mathbf{V}_{2r}\mathbf{V}_{2r}^T)\|_F^2 \leq \|\mathbf{A}_{(2r-1)} - (\mathbf{A}_{(2r-1)})_k\|_F^2 + \epsilon\|\mathbf{A}_{(2r-1)}\|_F^2 \tag{A.55}$$

As discussed in the analysis of Algorithm 2, observe that computing the $\mathrm{SVD}$ or even running an input-sparsity time algorithm as we recurse back up to the top becomes prohibitively expensive and no longer sublinear. Therefore, we follow the previous strategy of setting up a regression problem, sketching it and solving it approximately using leverage scores. We observe that an approximately optimal solution for $\mathbf{A}_{(2r-1)}$ lies in the row space of $\mathbf{V}_{2r}^T$ and set up the following regression problem:

$$\min_{\mathbf{X}} \|\mathbf{A}_{(2r-1)} - \mathbf{X}\mathbf{V}_{2r}^T\|_F^2 \tag{A.56}$$

Though this problem is small and independent of $m$ and $n$, as we recurse up, the regression problems grow larger and larger. Therefore, we sketch it using the leverage scores of $\mathbf{V}_{2r}^T$. Note, since $\mathbf{V}_{2r}^T$ is orthonormal, the leverage scores are precomputed. With probability at least $98/100$, we compute $\mathbf{X}_{\mathbf{A}_{(2r-1)}} = \operatorname{argmin}_{\mathbf{X}} \|\mathbf{A}_{(2r-1)}\mathbf{E}_{2r-1} - \mathbf{X}\mathbf{V}_{2r}^T\mathbf{E}_{2r-1}\|$, where $\mathbf{E}_{2r-1}$ is a leverage score sketching matrix with $\mathrm{poly}\left(\frac{k}{\epsilon}\right)$ columns. Given the sketching guarantee of Theorem 4.3,

$$\begin{aligned}
\|\mathbf{A}_{(2r-1)} - \mathbf{X}_{\mathbf{A}_{(2r-1)}}\mathbf{V}_{2r}^T\|_F^2 &\leq (1+\epsilon)\min_{\mathbf{X}} \|\mathbf{A}_{(2r-1)} - \mathbf{X}\mathbf{V}_{2r}^T\|_F^2 \\
&\leq (1+\epsilon)\|\mathbf{A}_{(2r-1)} - \mathbf{A}_{(2r-1)}\mathbf{V}_{2r}\mathbf{V}_{2r}^T\|_F^2 \\
&\leq \|\mathbf{A}_{(2r-1)} - (\mathbf{A}_{(2r-1)})_k\|_F^2 + O(\epsilon)\|\mathbf{A}_{(2r-1)}\|_F^2
\end{aligned} \tag{A.57}$$

where the last inequality follows from equation A.55.Therefore, $\mathbf{X}_{\mathbf{A}_{(2r-1)}} \mathbf{V}_{2r}^T$ has rank at most $k$ and is an approximate optimal solution for $\mathbf{A}_{(2r-1)}$. Applying Lemma A.4 again, we know that projecting onto the column space of $\mathbf{X}_{\mathbf{A}_{(2r-1)}} \mathbf{V}_{2r}^T$ is an approximately optimal solution for $\mathbf{A}_{(2r-2)}$. Formally, let $\mathbf{X}_{\mathbf{A}_{(2r-1)}} \mathbf{V}_{2r}^T = \mathbf{U}_{2r-1} \mathbf{V}_{2r-1}$ such that $\mathbf{U}_{2r-1}$ has orthonormal columns. It follows from equation A.57 that $\|\mathbf{A}_{(2r-1)} - \mathbf{U}_{2r-1} \mathbf{V}_{2r-1}\|_F^2 = \|\mathbf{A}_{(2r-1)} - (\mathbf{A}_{(2r-1)})_k\|_F^2 \pm \epsilon\|\mathbf{A}_{(2r-1)}\|_F^2$. Therefore,

$$\|(\mathbb{I} - \mathbf{U}_{2r-1}\mathbf{U}_{2r-1}^T)\mathbf{A}_{(2r-2)}\|_F^2 \le \|\mathbf{A}_{(2r-2)} - (\mathbf{A}_{(2r-2)})_k\|_F^2 + \epsilon\|\mathbf{A}_{(2r-2)}\|_F^2 \qquad \text{(A.58)}$$

We observe that a good solution for $\mathbf{A}_{(2r-2)}$ exists in the column space of $\mathbf{U}_{2r-1}$, and set up the following regression problem:

$$\min_{\mathbf{X}} \|\mathbf{A}_{(2r-2)} - \mathbf{U}_{2r-1}\mathbf{X}\|_F^2 \qquad \text{(A.59)}$$

Again, we sketch this regression problem using the leverage scores of $\mathbf{U}_{2r-1}$. Recall, $\mathbf{U}_{2r-1}$ has orthonormal rows and thus the leverage scores are precomputed. We can then compute $\mathbf{X}_{\mathbf{A}_{(2r-2)}} = \mathrm{argmin}_{\mathbf{X}}\|\mathbf{E}_{2r-2}\mathbf{A}_{(2r-2)} - \mathbf{E}_{2r-2}\mathbf{U}_{2r-1}\mathbf{X}\|_F^2$, where $\mathbf{E}_{2r-2}$ is a leverage score sketching matrix with $\mathrm{poly}(\frac{k}{\epsilon})$ rows. Given the sketching guarantee of leverage score sampling in Theorem 4.3,

$$\begin{aligned}
\|\mathbf{A}_{(2r-2)} - \mathbf{U}_{2r-1}\mathbf{X}_{\mathbf{A}_{(2r-2)}}\|_F^2 &\le (1+\epsilon)\min_{\mathbf{X}}\|\mathbf{A}_{(2r-2)} - \mathbf{U}_{2r-1}\mathbf{X}\|_F^2 \\
&\le (1+\epsilon)\|\mathbf{A}_{(2r-2)} - \mathbf{U}_{2r-1}\mathbf{U}_{2r-1}^T\mathbf{A}_{(2r-2)}\|_F^2 \qquad \text{(A.60)}\\
&\le \|\mathbf{A}_{(2r-2)} - (\mathbf{A}_{(2r-2)})_k\|_F^2 + O(\epsilon)\|\mathbf{A}_{(2r-2)}\|_F^2
\end{aligned}$$

We observe that $\mathbf{U}_{2r-1}\mathbf{X}_{\mathbf{A}_{(2r-2)}}$ is a rank-$k$ matrix, written in factored from, that is an approximate solution for $\mathbf{A}_{(2r-2)}$. Using $\mathbf{U}_{2r-1}, \mathbf{X}_{\mathbf{A}_{(2r-2)}}$, we can repeat the above analysis $r$ times all the way up the recursion stack. Note, the last level of the analysis is simply the one presented in the proof of Theorem 4.1. Let $\mathbf{U}_1, \mathbf{X}_{\mathbf{A}}$ be the solution obtain from solving the final regression problem. Note, union bounding over all the random events above, we obtain $\mathbf{U}_1, \mathbf{X}_{\mathbf{A}}$ with probability at least $9/10$. Setting $\mathbf{M} = \mathbf{U}_1$ and $\mathbf{N}^T = \mathbf{X}_{\mathbf{A}}$ finishes the proof and satisfied the *additive error* guarantee for $\mathbf{A}$. $\qquad \square$

Next, we show the running time of Algorithm 3 is sublinear in $m$ and $n$ for an appropriate setting of $b_1$, $b_2$ and $\gamma$.

**Lemma A.6.** *Let* $\mathbf{A} \in \mathbb{R}^{m\times n}$ *be a matrix such that it satisfies approximate triangle inequality. Then, for any* $\epsilon > 0$, *integer* $k$, *and a small constant* $\gamma > 0$, *Algorithm 3 runs in* $\widetilde{O}\left((m^{1+\gamma} + n^{1+\gamma})\,\mathrm{poly}(\frac{k}{\epsilon})\right)$ *time.*

*Proof.* Recall, from the running time analysis of Algorithm 2, for a $t_i \times s_i$ matrix $\mathbf{A}_{(i)}$, we a construct *column and row pcp* in $O\left(\frac{t_i \log(t_i)}{b_1}\mathrm{poly}(\frac{k}{\epsilon}) + b_1 s_i + t_i\right)$ and $O\left(\frac{s_i \log(s_i)}{b_2}\mathrm{poly}(\frac{k}{\epsilon}) + (b_2 t_i + s_i)\right)$ respectively. Since $i \in [2r]$ and $2r = O(1/\gamma)$, we can compute $\mathbf{A}_{(2r)}$ in $O\left((b_1 n + b_2 m)\,\mathrm{poly}(\frac{k\log(mn)}{\epsilon})\right)$, since the running time is dominated by sampling $b_1$ entries in each column and $b_2$ entries in each row of the input matrix $\mathbf{A}$, at the top level. Note, $\mathbf{A}_{(2r)}$ is a $t_{2r} \times s_{2r}$ matrix, where $t_{2r} = \frac{m}{b_2^{1/\gamma}}\mathrm{poly}(\frac{k\log(m)}{\epsilon})$ and $s_{2r} = \frac{n}{b_1^{1/\gamma}}\mathrm{poly}(\frac{k\log(n)}{\epsilon})$. We can compute the SVD of $\mathbf{A}_{(2r)}$ in $O\left(\left(\frac{nm}{(b_1 b_2)^{1/\gamma}}\right)^2 \mathrm{poly}(\frac{k\log(m)\log(m)}{\epsilon})\right)$. Next, we solve $2r$ regression problems by sketching them as we recurse back up. We again upper bound each recursive step by the running time of the top level. Recall, from the analysis of Algorithm 2, the regression problem can be solved in $O\left((m+n)\mathrm{poly}(\frac{k\log(m)\log(n)}{\epsilon})\right)$. Setting $b_1 = n^\gamma$ and $b_2 = m^\gamma$, the overall running time is $\widetilde{O}\left((m^{1+\gamma} + n^{1+\gamma})\,\mathrm{poly}(\frac{k}{\epsilon})\right)$. $\qquad \square$

Lemma $A.5$ together with Lemma $A.6$ imply our main theorem:

**Theorem A.3.** *(Sublinear Low-Rank Approximation for Distance Matrices.) Let* $\mathbf{A} \in \mathbb{R}^{m\times n}$ *be a matrix such that it satisfies approximate triangle inequality. Then, for any* $\epsilon > 0$, *integer* $k$ *and a small constant* $\gamma > 0$, *there exists an algorithm that accesses* $O\left(m^{1+\gamma} + n^{1+\gamma}\right)$ *entries of* $\mathbf{A}$ *and*

*runs in time* $\widetilde{O}\left(\left(m^{1+\gamma} + n^{1+\gamma}\right) poly(\frac{k}{\epsilon})\right)$ *to output matrices* $\mathbf{M} \in \mathbb{R}^{m \times k}$ *and* $\mathbf{N} \in \mathbb{R}^{n \times k}$ *such that with probability at least* $9/10$,

$$\|\mathbf{A} - \mathbf{M}\mathbf{N}^T\|_F^2 \leq \|\mathbf{A} - \mathbf{A}_k\|_F^2 + \epsilon\|\mathbf{A}\|_F^2$$

### A.7 Relative Error Guarantees

**Theorem 5.1.** *(Lower bound.) Let* $\mathbf{A}$ *be an* $n \times n$ *distance matrix. Let* $\mathbf{B}$ *be a rank-poly$(k)$ matrix such that* $\|\mathbf{A} - \mathbf{B}\|_F^2 \leq c\|\mathbf{A} - \mathbf{A}_k\|$ *for any constant* $c > 1$. *Then, any algorithm that outputs such a* $\mathbf{B}$ *requires* $\Omega(nnz(\mathbf{A}))$ *time.*

*Proof.* Let $P = \{e_1, e_2 \ldots e_n\}$ be a set of $n$ standard unit vectors and $Q$ be a set of $n-1$ zero vectors along with one point $q$ such that $q = \pm e_i$, where $i$ and the sign are chosen uniformly at random. Let the underlying metric space be $\ell_\infty$-norm. Note, all but one pairwise distances in $\mathbf{A}$ are 1. Further, one entry in the $i^{th}$ row of $\mathbf{A}$ is either 2 or 0. Note, $\mathbf{A}$ is a rank-2 matrix and thus $\|\mathbf{A} - \mathbf{A}_k\|_F^2 = 0$, for all $k > 1$. Let $\mathbf{B}$ be a rank-poly$(k)$ matrix that obtains any relative-error guarantee. Then, $\mathbf{B}$ must exactly recover $\mathbf{A}$ and therefore any algorithm needs to read all entries of $\mathbf{A}$ in order to find the entry that is 0 or 2. $\square$

---

**Algorithm 1** Bi-criteria Algorithm for Euclidean Matrices.

---

**Require:** Euclidean Matrix $\mathbf{A}_{n \times n}$, integer $k$.
1: Let $\mathbf{A} = \mathbf{A}_1 + \mathbf{A}_2 - 2\mathbf{B}$ s.t. $\mathbf{A}_1$ and $\mathbf{A}_2$ are rank-1 matrices and $\mathbf{B}$ is a PSD Matrix.
2: Then, $\mathbf{A}_1 = \mathbf{a}_1\mathbf{a}_1^T$ and $\mathbf{A}_2 = \mathbf{a}_2\mathbf{a}_2^T$.
3: Compute $\mathbf{M}\mathbf{N}^T$ by running the sublinear low-rank approximation algorithm from [14] on $\mathbf{B}$ with parameter $k + 2$.
4: Compute $\mathbf{V}$ an orthonormal basis for $\mathbf{M}$.
5: Let $\mathbf{W}$ be $\mathbf{V}$ concatenated with $\mathbf{a}_1$ and $\mathbf{a}_2$. We denote $\mathbf{W}$ as $[\mathbf{V}; \mathbf{a}_1, \mathbf{a}_2]$.
**Ensure:** $\mathbf{W}\mathbf{W}^T$

---

**Theorem 5.2.** *(Bi-criteria Algorithm.) Let* $\mathbf{A}$ *be a Euclidean Distance matrix. Then, for any* $\epsilon > 0$ *and integer* $k$, *with probability at least* $9/10$, *Algorithm 1 outputs a rank* $(k+4)$ *matrix* $\mathbf{W}\mathbf{W}^T$ *such that*

$$\|\mathbf{A} - \mathbf{A}\mathbf{W}\mathbf{W}^T\|_F \leq (1 + \epsilon)\|\mathbf{A} - \mathbf{A}_k\|_F$$

*where* $\mathbf{A}_k$ *is the best rank-k approximation to* $\mathbf{A}$. *Further, Algorithm 1 runs in* $O(npoly(\frac{k}{\epsilon}))$.

*Proof.* First, we show that we can simulate the sublinear PSD algorithm from [14] on $\mathbf{B}$, given random access to $\mathbf{A}$ and reading $n-1$ additional entries. Observe that computing $\mathbf{B}_{i,j}$ requires $\|x_i\|_2^2$ and $\|x_j\|_2^2$. Since pairwise distances are invariant to a uniform shift in position. Therefore, w.l.o.g. we can assume that the first point, $x_1$, is at the origin. Then, the $j^{th}$ entry of the first row $\mathbf{A}$ is $\|x_j\|_2^2$. Now, $\mathbf{B}_{i,j} = (\|x_i\|_2^2 + \|x_j\|_2^2 - \mathbf{A}_{i,j})/2$ and we have access to each of these values. Therefore, we can simulate the algorithm for sublinear low-rank approximation on $\mathbf{B}$ in $O(npoly(\frac{k}{\epsilon}))$. Note, in Algorithm 1 we find a rank $k + 1$ approximation to $\mathbf{B}$ and by Theorem **??**, the algorithm outputs matrices $\mathbf{M}, \mathbf{N} \in \mathbb{R}^{n \times k}$ such that

$$\|\mathbf{B}(\mathbb{I} - \mathbf{M}\mathbf{N}^T)\|_F \leq (1 + \epsilon)\|\mathbf{B} - \mathbf{B}_{k+2}\|_F$$

where $\mathbf{B}_{k+2}$ is the best rank-$(k + 2)$ approximation to $\mathbf{B}$. Let $\mathbf{V}$ be an orthonormal basis for $\mathbf{M}$, which can be computed in $O(nk^2)$ time. Let $\mathbf{W} = [\mathbf{V}; \mathbf{a}_1, \mathbf{a}_2]$. We observe that the projection matrix $\mathbf{W}\mathbf{W}^T$ applied to $\mathbf{A}_1$ and $\mathbf{A}_2$ yields $\mathbf{A}_1$ and $\mathbf{A}_2$ respectively since they lie in the column space of $\mathbf{W}$. Therefore,

$$
\begin{aligned}
\|\mathbf{A}(\mathbb{I} - \mathbf{W}\mathbf{W}^T)\|_F &= \|(\mathbf{A}_1 + \mathbf{A}_2 - 2\mathbf{B})(\mathbb{I} - \mathbf{W}\mathbf{W}^T)\|_F \\
&= \|(\mathbf{A}_1 - \mathbf{A}_1\mathbf{W}\mathbf{W}^T) + (\mathbf{A}_2 - \mathbf{A}_2\mathbf{W}\mathbf{W}^T) - 2(\mathbf{B} - \mathbf{B}\mathbf{W}\mathbf{W}^T)\|_F \\
&= 2\|\mathbf{B}(\mathbb{I} - \mathbf{W}\mathbf{W}^T)\|_F \qquad\qquad\qquad\qquad\qquad\text{(A.61)} \\
&\leq 2\|\mathbf{B}(\mathbb{I} - \mathbf{M}\mathbf{N}^T)\|_F \\
&\leq 2(1 + \epsilon)\|\mathbf{B} - \mathbf{B}_{k+2}\|_F
\end{aligned}
$$

| Metric | SVD | IS | Sub | Iter |
|--------|--------|-------|------|-------|
| $L_2$ | 473.83 | 38.04 | 2.09 | 21.53 |
| $L_1$ | 526.11 | 37.62 | 1.92 | 20.26 |
| $L_\infty$ | 417.42 | 38.40 | 1.93 | 27.66 |
| $L_c$ | 476.89 | 40.95 | 1.93 | 22.53 |

Table 3: Running Time (in seconds) of full SVD, Input-sparsity, Sublinear and Iterative SVD algorithms on the Gisette Dataset for Rank = 40

| Metric | SVD | IS | Sub | Iter |
|--------|--------|-------|------|-------|
| $L_2$ | 480.19 | 39.69 | 2.86 | 16.37 |
| $L_1$ | 417.35 | 37.15 | 3.04 | 16.09 |
| $L_\infty$ | 447.39 | 41.89 | 3.37 | 17.68 |
| $L_c$ | 425.95 | 45.99 | 2.41 | 20.12 |

Table 4: Running Time (in seconds) of full SVD, Input-sparsity, Sublinear and Iterative SVD algorithms on the Poker Dataset for Rank = 40.

Next, we bound $\|\mathbf{A} - \mathbf{A}_k\|_F$ in terms of $\|\mathbf{B} - \mathbf{B}_{k+2}\|_F$ as follows:

$$\|\mathbf{A} - \mathbf{A}_k\|_F = \|\mathbf{A}_1 + \mathbf{A}_2 - 2\mathbf{B} - \mathbf{A}_k\|_F$$
$$= 2\left\|\mathbf{B} - \frac{(\mathbf{A}_1 + \mathbf{A}_2 - \mathbf{A}_k)}{2}\right\|_F \qquad \text{(A.62)}$$

Observe, $\mathbf{A}_1 + \mathbf{A}_2 - \mathbf{A}_k$ is a rank-2 perturbation to a rank-$k$ matrix, therefore, $\left\|\mathbf{B} - \frac{(\mathbf{A}_1 + \mathbf{A}_2 - \mathbf{A}_k)}{2}\right\|_F \geq \|\mathbf{B} - \mathbf{B}_{k+2}\|_F$. Combining the two equations, we get $\|\mathbf{A}(\mathbb{I} - \mathbf{WW}^T)\|_F \leq (1 + \epsilon)\|\mathbf{A} - \mathbf{A}_k\|_F$. Recall, $\mathbf{W} = [\mathbf{V}; \mathbf{a}_1, \mathbf{a}_2]$, where $\mathbf{V}$ has rank $k + 2$ and thus $\mathbf{W}$ has rank at most $k + 4$. $\qquad\qquad\square$

## B Experiments

In this section, we present the results of our sublinear time algorithm, the conventional SVD Algorithm (optimal error), iterative SVD methods and the input-sparsity time algorithm from [4] on the Gisette and Poker Dataset. Note, we don't include the error of the Iterative SVD algorithm in our plots since it is orders of magnitude larger than the SVD error and makes the plots less interpretable. For instance, on the Gisette dataset the minimum Frobenius norm error attain by SVD is $1086.40$, whereas that of the Iterative SVD is $26156230.14$.

## Gisette Dataset

### Euclidean Distance

### Manhattan Distance

### Chebyshev Distance

### Canberra Distance

## Poker Dataset

### Euclidean Distance

### Manhattan Distance

### Chebyshev Distance

### Canberra Distance

Figure B.1: Here we consider the Gisette dataset [11] and the Poker Dataset [2]. The distance matrix is created using Euclidean, Manhattan, Chebyshev and Canberra distance metrics. We compare the error achieved by SVD (optimal), our Sublinear Algorithm and an Input Sparsity Algorithm.