[Reviews · NeurIPS 2018]

Reviewer 1



This paper presents an algorithm for computing a near-optimal low-rank approximation for a distance matrix — i.e. a matrix where the (i,j)th entry equals d(p_i, q_j) for some metric d and sets of points p_1, …, p_m and q_1,… q_n. Distance matrices are common in learning and data analysis tasks and low-rank approximation is a ubiquitous approach to compressing matrices. In general, the fastest existing low-rank approximation algorithms are randomized and, to compute a k-rank approximation, run in O(nnz(A) + min(n,m)*poly(k/eps)) time for a matrix A with nnz(A) non-zeros. Since the second term is often considered lower order (k is typically << n and eps is moderate), these algorithms run in roughly linear time in the size of the input. Naturally, this is optimal for general matrices — if I don’t at least read every entry in the matrix, I could miss a very heavy entry that I would need to see to output a good low-rank approximation. However, we could ask if better runtimes are possible for matrices with more structure, like distance matrices. We might hope for algorithms that run in sublinear time. Recent work shows that, surprisingly, sublinear time low-rank approximation algorithms can be obtained for positive semidefinite matrices, another structured class of matrices. This result relies on the fact that it’s not possible to add an arbitrarily heavy element A_i,j to a PSD matrix without increasing the size of either A_i,i or A_j,j. Accordingly, simply reading the diagonal of A (which only takes n time) avoids the trivial lower bounds that apply to general matrices. The authors begin with the observation that a similar statement is true for distance matrices. By triangle inequality, if A_i,j is very heavy, A_1,i, A_1,j, or A_1,1 must be too. These entries can all be found by reading the first row of A (or really any single row of A_. As in the case of PSD matrices, this property allows us to avoid lower bounds for general matrices. The authors leverage this observation to motivate their algorithm, which is based on solving the low-rank approximation using a random subsample of the original matrix. In particular, they sample rows/columns by their norms. It is well known that subsampling in this way makes it possible to obtain a low-rank approximation within additive error eps*||A||_F^2 of optimal. For general matrices, computing norms naively takes nnz(A) time. However, the authors show that for distance matrices, norms can be estimated coarsely in sublinear time. They can subsample the matrix using these coarse estimates, and then recurse to shrink the problem size even smaller. In recursing, the authors alternate between row and column sampling. Overall, this yields an algorithm that runs in (min(N,m)^1+gamma) * poly(k/eps)^1/gamma time for any chosen gamma, which controls the levels of recursion in the algorithm. The authors present a simpler algorithm with just one level of recursion that runs in min(n,m)^1.34 * poly(k/eps) time. The authors implement this algorithm and it turns out to be quite practical, outperforming nnz(A) time random projection methods pretty significantly. In addition to their main result, the authors include a number of additional results. Of particular interest is a lower bound, which demonstrates that additive ||A||_F^2 error is necessary — achieving full relative error for the low-rank approximation problem isn’t possible even for distance matrices without at least reading the whole matrix. This places the complexity of low-rank approximation for distances matrices somewhere between standard matrices, where even additive error is not possible in sublinear time, and PSD matrices, where full relative error is possible in sublinear time. Overall, I think the paper is well written and the techniques are new and interesting. It’s also nice to see that the main algorithm presented is already practical. One concern with the paper is that I find the proof of the faster (min(N,m)^1+gamma) * poly(k/eps)^1/gamma difficult to follow. In particular, recursive sampling cannot be performed naively because, after sampling and reweighting rows from a distance matrix, you end up with a matrix that is no longer a distance matrix. To deal with this challenge, the authors suggest partitioning sampled rows into “buckets” based on how much they were reweighed by. The matrix to be recursed on is a valid distance matrix, at least when restricted to rows in one bucket, which allows column norms to be estimated by estimating the norm in each block of rows separately (estimating column norms is need for the next step of reduction). This makes perfect sense to me for the two step in(n,m)^1.34 * poly(k/eps) time algorithm (Algorithm 3), but I do not understand how the approach works for the faster Algorithm 2. In particular, I don’t see how the argument carries through multiple levels of recursion without blowing up the number of buckets and, for whatever reason, Algorithm 2 does not include the reweighing step in its pseudocode! I would encourage the authors to add it in explicitly. If this makes the algorithm much more difficult to present, I would encourage the authors to present the simpler, but still interesting Algorithm 3 in the main part of their paper, and to give a full and rigorous development of Algorithm 2 in the appendix.

Reviewer 2



The paper gives a sublinear time algorithm to obtain a low rank approximation of a distance matrix A. (Entry (i,j) is the distance between p_i and q_j). The intuition is that because A is a distance matrix you do not have to look at all its entries to get a good low rank approximation: The triangle inequality imposes some structure on the entries that you can use and look at only a few. The main idea is to fast estimate the norms of the columns and the rows, use these estimates for a weighted sampling of the rows and the columns such that the resulting submatrix "preserves the projection cost" on low rank subspaces. The final algorithm is complicated and I must admit that I could not fully absorb what it does even after investing a substantial amount of time. It may be better to put first the simpler scheme (the one that you implemented) which is in the appendix right now and not the most general scheme (replace the order of presentation between Algorithm 3 and Algorithm 2). The description of the main result is heavy and confusing and can be improved a lot. For example, I see clustering into weight classes only in Algorithm 3, but you talk about this in the main text before Algorithm 2 (line 193). You do not make clear the exact differences between the algorithms, why the come about, and what are their consequences. I think you can make a substantially better job in making this more accessible. minor comments: 1) The text in lines 119-130 is confusing. Better to write an outline of the proof. I only could get it by reading the proof. For example diam(Q) is not defined ? you have strange factors of "4" and "2" that are not clear, and its not clear that you cover all cases. Don't use a different notation for diam(Q) in the appendix, it may be confusing. 2) line 183 "minimizes" over what ? (say that you minimize over X explicitly) 3) line 192: "stem" ==> step 4) line 380, 381 why do you write A_{x,j} twice in the displays 5) line 445: what is the Matrix Bernstein inequality ?

Reviewer 3



This paper considers the problem of computing low rank approximations to a pairwise distance matrix in sublinear time where each entry (i,j) is the distance between some element p_i and p_j according to some distance metric. While computing such an approximation in sublinear time is impossible for general matrices, they show that for distance matrices additive approximations are always computable in sublinear time, or more precisely time linear in the number of elements and polynomial in the desired rank and accuracy (which is sublinear as if the elements are distinct then the distance matrix is dense). Furthermore, they provide empirical evidence that their method runs fast in practice, show that computing relative approximations in sublinear time is impossible in general, and show how to do this for matrices that contain the squares of Euclidean distances. To achieve this result they provide a procedure for achieving non-trivial approximations to the row and column norms of a distance matrix in sublinear time. This procedure work simply by looking at a single row and column of the matrix and then uniform sampling from all rows. Furthermore, they show that having such a procedure suffices to recursively subsample the matrix, ultimately computing a low rank approximation of the matrix. The work builds off previous work on computing sublinear time low rank approximations of positive semidefinite (PSD) matrices and leverages a variety of recent result on fast linear algebraic subroutines (e.g. input sparsity regression). This is a nice result that makes substantial progress on the fundamental problems of low rank approximation and efficiently processing distance data. Moreover, they show that their algorithms are robust to matrices containing distance-like information that only approximately satisfies triangle inequality (and leverage this in their algorithm), thus making their results of even broader utility for processing more data sets. That they achieve their results through a general using norm approximations also seems to be of greater utility. That said, the writing in the paper could use some improvement. Having a preliminary section that clearly states their notation might help and clearly specifying what notion of approximation is referred to (e.g. Lemma 2.1) would help the readability of the paper. Also, better specifying how to think of the trade-offs in using the lemmas or what parameter regime the lemmas in Section 3 would be applied to would help. Also, comments like, “by dividing subsampled matrices” in Page 5 could use further clarification. Furthermore, more discussion of exactly the relationship of this paper to the approach in MW17 for approximating PSD matrices and what was known about computing low rank approximations with a norm-approximating oracle, would better clarify the context and contributions of this paper. However, the above concerns are relatively minor given the interest and utility of the central results in this paper and therefore I recommend acceptance. Typos / Minor Writing Comments: - Page 3: how is diam defined if distances within Q are not given by the distance matrix. - Page 4: saying uniform sampling suffices seems misleading as computing d is also necessary - Page 4: missing some bolding of A - Page 5: stem --> step EDIT: Thank you for the thoughtful rebuttal. I encourage you to add the information you provided there to the paper.